# The Arp2/3 complex is required for in situ haptotactic response of microglia to iC3b

Summer G Paulson [1,2], Isabella Swafford[1,2], Fritz W Lischka [2,3,4] & Jeremy D Rotty [1✉]

## Abstract

Microglia maintain brain homeostasis via iC3b-mediated synaptic pruning. The Arp2/3 complex has been implicated in iC3b-mediated macrophage phagocytosis, but it is unclear whether it is similarly required in microglia in the CNS. We examined the question of CR3-dependent clearance of iC3b in microglia using a combination of in vitro and in situ physical confinement studies. Arp2/3 inhibition decreased iC3b phagocytosis and cell motility in vitro. Furthermore, microglia-like cells remove immobilized iC3b from the substrate in an Arp2/3-dependent fashion, in a process reminiscent of trogocytic synaptic pruning. We also used a novel approach to immobilize an iC3b gradient onto a substrate and demonstrate Arp2/3-dependent haptotactic migration toward increasing iC3b concentrations. While Arp2/3-deficient microglia robustly respond to ATP via chemotaxis within mouse hippocampal slices, they demonstrate a persistent inability to stably interact with iC3b-coated beads. The present study establishes new approaches to systematically interrogate molecular pathways relevant to synaptic pruning, advances the understanding of iC3b phagocytosis as a haptotactic response, and confirms that the Arp2/3-dependent haptotactic response is important for microglia function in the CNS microenvironment.

Keywords Microglia; iC3b Phagocytosis; Arp2/3 Complex; Haptotaxis; Synaptic Pruning
Subject Categories Cell Adhesion, Polarity & Cytoskeleton; Membranes & Trafficking; Neuroscience

## Introduction

Microglia are the resident immune cells of the central nervous system (CNS) (Brabazon et al, 2018; Melo et al, 2022). They respond to a variety of phagocytic markers, including the complement proteins iC3b and C1q, that label phagocytic targets upon activation of the complement cascade (Janeway et al, 2001). These factors clear foreign bodies and debris but also have been implicated in pruning dendritic spines in the CNS (Warwick et al, 2021; Dunkelberger and Song, 2010;

Dalakas et al, 2020). Dendritic spines, located along the length of neuronal dendrites, are postsynaptic sites that facilitate signal relay between neurons (Kim et al, 2013b). During development, neurons initially overpopulate their dendrites with new dendritic spines (Runge et al, 2020). Excessive spines are then marked for microglial phagocytosis with iC3b (Khanal and Hotulainen, 2021). Microglia respond to the iC3b cue via complement receptor 3 (CR3, also known as Mac1) (Gordon et al, 1987). This results in synaptic pruning and remodeling of the neuronal network by microglia (Schafer et al, 2012).

The CR3 receptor is composed of 2 integrins: CD11b (αM integrin) and CD18 (β2 integrin) (Zhang et al, 2017). The αMβ2 heterodimer binds directly to phagocytic cues such as iC3b and initiates the process of internalization via mechanosensitive clutch formation and Arp2/3 complex-dependent membrane protrusion (Pang et al, 2023). Integrins are also vital in adhesion-dependent mesenchymal cell migration by similarly linking the surrounding extracellular matrix (ECM) to adhesive structures and the actomyosin machinery in cells (Jaumouillé et al, 2019). The seven-subunit Arp2/3 complex generates dense branched actin networks in response to integrin engagement in the context of migration and phagocytosis (Helgeson and Nolen, 2013). Upon docking to a pre-existing 'mother' actin filament, Arp2/3 complex polymerizes a branched 'daughter' filament at a characteristic 70° angle (Rotty et al, 2013). The growing barbed (+) ends of these branched actin arrays provide the physical force that pushes against the membrane, causing characteristic dynamic protrusions at the cell edge called lamellipodia (Rotty et al, 2013). Arp2/3 complex-deficient cells fail to generate iC3b-responsive phagocytic cups and cannot sense fibronectin gradients, suggesting a fundamental defect related to integrin sensing (Rotty et al, 2017). However, it is not yet known whether the Arp2/3 complex is necessary for complement-mediated phagocytosis in vivo, such as in the CNS during synaptic pruning.

Complement-mediated phagocytosis has been studied extensively in macrophages, examining phagocytic cup formation, macrophage response to complement signaling, and downstream cytokine production (Acharya et al, 2020; Walbaum et al, 2021). However, much of this work has been done in immortalized macrophage lines in vitro in 2D culture experiments that do not consider how tissue confinement or interaction with the complex set of cues in the endogenous microenvironment might modulate this process. Furthermore, insights generated from studying macrophage complement-mediated phagocytosis may not directly translate to microglia as these cell types are vastly different, despite microglia commonly being referred to as the macrophages of the central nervous system. A study

[1]Uniformed Services University of the Health Sciences, Department of Biochemistry, Bethesda, MD 20814, USA. [2]The Henry M. Jackson Foundation for the advancement of Military Medicine, Bethesda, MD 20817, USA. [3]Uniformed Services University of the Health Sciences, Department of Anatomy, Physiology, and Genetics, Bethesda, MD 20814, USA. [4]Uniformed Services University of the Health Sciences, Biomedical Instrumentation Center, Bethesda, MD 20814, USA. ✉E-mail: Jeremy.Rotty@usuhs.edu

   

examining gene expression differences between microglia and monocyte-derived macrophages found 239 genes specifically expressed in microglia (Butovsky et al, 2014). In addition, microglia have a dramatically different morphology and in vivo behavior compared to tissue macrophages (Butovsky et al, 2014). Prior research on the Arp2/3 complex within the CNS has focused on how neuronal Arp2/3 complex supports dendritic spine generation and morphology (Kim et al, 2013a). The role of the Arp2/3 complex within microglia, and how it responds to specific environmental cues remains understudied in comparison.

In the present work, we use a combination of in vitro and in situ physical confinement studies to interrogate CR3-dependent clearance of iC3b in microglia. Arp2/3 complex inhibition decreased phagocytosis and cell motility in vitro. Furthermore, we demonstrate that microglia-like cells are able to remove immobilized iC3b from the substrate in an Arp2/3-dependent fashion, in a process reminiscent of trogocytic synaptic pruning. Trogocytosis is the act of one cell phagocytizing only a small portion of another cells, differing from phagocytosis due to the way engulfment occurs (Miyake and Karasuyama, 2021; Watanabe et al, 2020). We also used a novel approach to immobilize an iC3b gradient onto a substrate and demonstrate Arp2/3-dependent haptotactic migration toward increasing iC3b concentrations. Finally, we quantified microglial response to ATP and iC3b-labeled beads in murine hippocampal brain slices to interrogate extracellular sensing within the physiological microenvironment. In line with a significant body of in vitro work, we demonstrate that Arp2/3 complex inhibition does not impact the microglial response to the chemotactic ligand ATP. However, microglia demonstrate a persistent inability to stably interact with iC3b-coated beads upon Arp2/3 complex inhibition. Taken together, the present study establishes new approaches to systematically interrogate molecular pathways relevant to synaptic pruning, advances the understanding of iC3b phagocytosis as a haptotactic response, and confirms that the Arp2/3-dependent haptotactic response is relevant for microglia in their normal physiological microenvironment. The importance of the Arp2/3 complex for microglia function is further underlined by work by Safaiyan and co-authors (Safaiyan et al, 2026), which convincingly demonstrates that microglia-restricted genetic deletion of the Arp2/3 complex disrupts microglial development and has non-autonomous consequences for neural homeostasis. Both studies demonstrate that the Arp2/3 complex contributes significantly to microglial function, and that its disruption may dramatically impact CNS health during aging.

# Results

## iC3b acts as a phagocytic cue independent of how cells interact with it

iC3b, a phagocytic "eat me" cue common to the brain and CNS, represents a microenvironmental signal that stimulates a specific microglial response in the context of synaptic pruning. To facilitate our assessment of microglia-iC3b interactions, we labeled purified human iC3b with Alexa Fluor 555 (denoted as AF-iC3b). During pilot studies using the iC3b to coat dishes, we noticed depositions of fluorescent puncta in the AF-iC3b wells. We confirmed that these puncta contained iC3b and not aggregated fluorescent dye by

demonstrating that the fluorescent signal correlated tightly with purified iC3b (Fig. EV1A). To confirm the specificity of the AF-iC3b interaction with CR3 in our cells, we performed adhesion assays after inhibiting the CR3 receptor using blocking antibodies against both integrin partners. BV-2 cells (a microglia-like cell line) were incubated with FBS (negative control), IgG (a negative isotype control), αM-, or β2-blocking antibodies before seeding onto either 10 μg/mL fibronectin (FN) or 20 μg/mL AF-iC3b-coated wells. After waiting 2 h for cells to adhere, each well was washed and imaged (Fig. EV1B). A significant reduction in adhesion to AF-iC3b occurred when either αM or β2 integrin was inhibited, while adhesion to FN remained unchanged by CR3 inhibition (Fig. EV1C,D). Additionally, there was no significant difference in the total number of cells present before the wash when comparing within antibody plating groups (Fig. EV1E). These findings demonstrate that AF-iC3b effectively coats glass and plastic, and that its interaction with our cells is CR3-dependent.

Previous research in the lab examined how different ECM coatings affect cell migration (Stinson et al, 2024). We hypothesized that iC3b similarly induces a migratory response. BV-2 cells were plated on 10 μg/mL FN or on 1 μg/mL, 10 μg/mL, or 20 μg/mL of AF-iC3b. Compared to FN, cells migrated slowly on AF-iC3b (Fig. 1A), with a dose-dependent decrease in accumulated distance traveled across the AF-iC3b concentrations (Fig. 1B). Lastly, cells plated on the AF-iC3b were less persistent, following the overall trend demonstrated by velocity measurements (Fig. 1C). Together these results suggest that microglia adhere more strongly to AF-iC3b than to FN. Therefore, we decided to examine whether this could be explained by cells removing AF-iC3b from the surface of the well rather than using it primarily as a migration substrate.

Creating binary masks and measuring the raw integrated density of the entire field of view, we examined differences between the beginning and end of the 16-h run to determine how much AF-iC3b had been removed. We observed a large amount of the coating was phagocytized, regardless of the surface concentration of AF-iC3b (Fig. 1D; Movie EV1, quantified in Fig. 1E). Cells became fluorescent as they ingested the AF-iC3b, demonstrating that they internalized the label. We also confirmed complement receptor specificity by treating with β2-blocking antibody and demonstrating a reduction in iC3b surface uptake, when plated on this substrate for 16 h (Fig. EV1F). The effect of αM blocking was not as dramatic, potentially because of compensation by CR4 (αXβ2). Next, we decided to explicitly investigate phagocytosis with pHrodo-iC3b-opsonized 2-micron polyspheres (here on referred to as beads).

The 2-micron bead size was chosen due to its similarity to the range of spine lengths measured experimentally (Benavides-Piccione et al, 2025; Nakayama et al, 2000; Hotulainen and Hoogenraad, 2010). The pHrodo tag fluoresces under acidic conditions, such as within the phagolysosome, allowing for unbiased detection of complete phagocytosis (Fig. EV2A). To rule out any signal from external beads, we compared external and internalized beads and demonstrated a clear difference in fluorescence between these two states (Fig. EV2B). Since modeling iC3b-dependent synaptic pruning is easily controlled in this in vitro environment, we decided to also incorporate confinement as an experimental factor in our phagocytosis assays, as the CNS is a highly confining environment. We utilized an under-agarose confinement system (Stinson et al, 2025), inspired by the work of Heit and Kubes (Heit and Kubes, 2003) (Fig. EV2A). With this assay, cells were either a) in wells filled with cell culture media

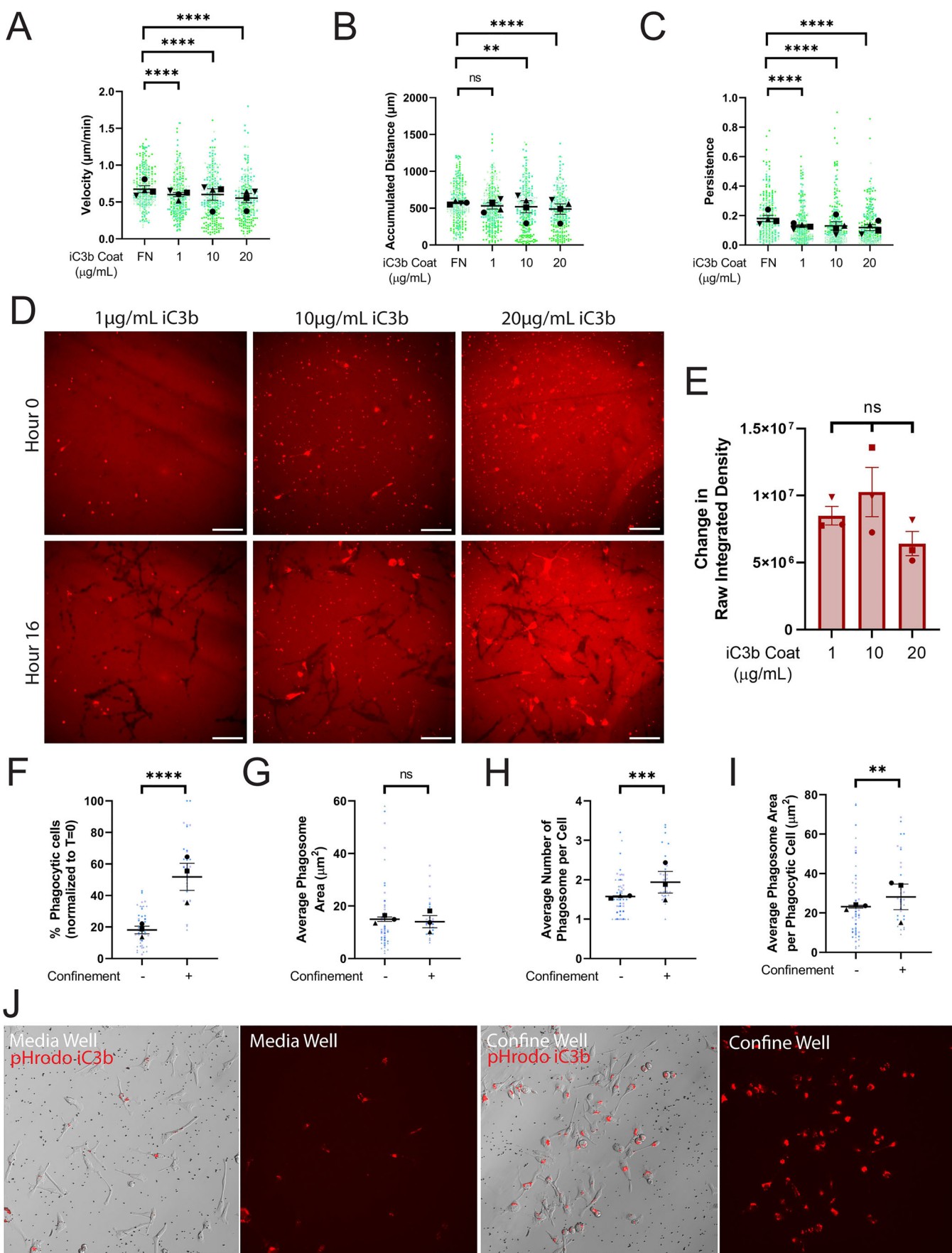

    This is a U.S. Government work and not under copyright protection in the US; foreign copyright protection may apply

◄  **Figure 1. iC3b phagocytosis is enhanced by confinement, not iC3b dosage.**

(A–E) Cells plated on either 10 μg/mL FN, 1 μg/mL AF-iC3b, 10 μg/mL AF-iC3b, or 20 μg/mL AF-iC3b. (A) Velocity of cells migrating (μm/min). (B) The maximum accumulated distance (μm) that the cell traveled. (C) Persistence of the cell during the length of its track. (D) Example images at each AF-iC3b coating dosage demonstrating phagocytosis of coating after 16 h. Scale bar represents 100 μm. (E) Quantification of amount of iC3b phagocytized from the bottom of the dish, as change in raw integrated density of fluorescence. (F–J) Cells plated on either in cell culture media or confined under 1% agarose and introduced to pHrodo-iC3b-opsonized beads. (F) The percentage of fluorescent cells in a field of view, comparing unconfined and confined cells, normalized to T = 0 h. Any cell fluorescent at T = 0 was removed from further counting. (G) The average phagosome area per phagocytic cell (μm²), calculated by dividing the average size of all phagosomes in a field of view by the number of fluorescent cells counted (denoted phagocytic cells). (H) Average number of phagosomes per phagocytic cell. (I) Average phagosome size (μm²). This is the average size of phagosomes per field of view divided by the average number of phagosomes per field of view. (J) Example composite images of phase contrast and pHrodo-red merged images 8 h after pHrodo-iC3b beads were added to media wells (left) or confined agarose wells (right). The pHrodo-red signal from these merged images is shown by itself alongside the merge to emphasize internalized pHrodo-iC3b-bead signal. Scale bar represents 100 μm. For all graphs, black points demonstrate experiment means, colored points demonstrate individual cell values for each run. N = 4 experiments (biological replicates) for migration (A–E) or N = 3 experiments (biological replicates) for phagocytosis (F–J); n = 50 cells for each condition per experiment for migration (A–C), n = 10 fields of view per condition for coating phagocytosis (E), or n = 30 fields of view for each condition per experiment for phagocytosis (F–J). For (A–E), statistical analysis was assessed using via Kruskal–Wallis test with Dunn multiple comparisons. For (F–J), statistical analysis was assessed using the Mann–Whitney tests. Error Bars represent SEM. Values for $p$ are as follows: (A) ****$p < 0.0001$. (B) ns nonsignificant, **$p = 0.0078$, ****$p < 0.0001$. (C) ****$p < 0.0001$. (E) ns nonsignificant. (F) ****$p < 0.0001$. (G) ns nonsignificant. (H) ***$p = 0.0001$. (I) **$p = 0.0051$. Source data are available online for this figure.

(Media Well) or b) confined underneath a layer of 1% agarose (Confined Well). Normalizing the number of beads in each condition was essential to rule out differences arising purely from altered local bead availability (Fig. EV2C,D). This is a factor primarily in interpreting under-agarose data, as beads injected under the agarose do not disperse evenly as they do in media. Due to this variability between confined fields of view, we calculated the number of beads in each field of view and categorized them into different density groups. As most media well fields of view were categorized as "low density" (Fig. EV2D, bottom), confined fields of view in the same category (Fig. EV2D, middle) were exclusively used for analysis.

Phagocytosis of pHrodo-iC3b-beads was measured at 2 h, before the cells reached bead saturation. Phagocytosis increased significantly under confinement, with a greater than twofold increase in the percentage of phagocytic cells compared to media wells (Fig. 1F; Movie EV2). This increase did not necessarily reflect an increase in overall bead consumption per cell, as indicated by average phagosome area remaining constant (Fig. 1G), though the number of phagosomes per phagocytic cell was increased under confinement (Fig. 1H). When dividing the total phagosome area in the field of view by the number of phagosomes in the field of view, we also observed no difference under confinement (Fig. 1I). This is also visible in Fig. 1J, where comparison of just the pHrodo images demonstrates similarly sized phagosomes for the media and confined wells. We once more used blocking antibodies to determine the complement receptor specificity of this response. As with our surface uptake experiments, bead internalization was substantially affected by β2 integrin blocking (Fig. EV1G). The effect of αM integrin blocking was not as pronounced, possibly due to compensation by αXβ2 as stated before (CR4). A recent study noted that BV-2 cells express CD11c (αX) in addition to CD11b (αM) (Iannucci et al, 2020). These data demonstrate that BV-2 cells interact with iC3b similarly, whether immobilized on the culture surface or on a latex bead, and that confinement enhances phagocytosis. Next, we wanted to interrogate whether the Arp2/3 complex was required for these responses.

**The Arp2/3 complex is necessary for iC3b phagocytosis**

The Arp2/3 complex has previously been implicated in iC3b-mediated phagocytosis (Rotty et al, 2017), fibronectin haptotaxis under confinement (Stinson et al, 2025) in macrophages. We hypothesized that inhibition of the Arp2/3 complex would similarly impair BV-2 phagocytosis and cell migration. We coated a 4-chamber dish with 10 μg/mL AF-iC3b and then added agarose to half of the wells. We also incorporated either DMSO vehicle control or the small molecular Arp2/3 complex inhibitor CK-666 into the media or agarose. Arp2/3 complex inhibition decreased migration velocity and accumulated distance compared to control, which was exacerbated further under confinement (Fig. 2A,B). Confinement alone did not shift these values, as both DMSO groups were similar to each other. However, confined DMSO-treated cells were more directionally persistent than unconfined DMSO-treated cells. (Fig. 2C). Notably, CK-666 treatment lowered unconfined migratory persistence compared to DMSO, and confinement was unable to rescue the persistence phenotype in Arp2/3 complex-deficient cells (Fig. 2C).

Interestingly, confinement does not significantly increase phagocytic uptake of surface-labeled AF-iC3b relative to unconfined cells in DMSO-treated conditions (Fig. 2D; Movie EV3, visualized in 2E). In contrast, confinement increased phagocytic uptake of iC3b-labeled beads (Fig. 1F). Arp2/3 complex inhibition decreased uptake of AF-iC3b compared to relevant DMSO-treated wells in unconfined and confined settings (Fig. 2D, visualized in 2E).

We then assessed whether the Arp2/3 complex was involved in the uptake of iC3b-beads. Therefore, we repeated our experimental parameters with CK-666 and confinement but introduced pHrodo-iC3b beads instead of AF-iC3b surface coating. Confinement failed to increase phagocytosis in the presence of DMSO, suggesting a modest suppressive effect (Fig. 2F) not observed when DMSO was absent as indicated by the higher overall phagocytic response in this setting (Fig. 1F). The presence of DMSO in the iC3b surface uptake experiment (Fig. 2D) might also explain why confinement didn't dramatically shift internalization in this context. DMSO interacts with sodium channels in CNS cells, prolonging the pro-inflammatory onset of cells that could suppress phagocytosis (Pappalardo et al, 2016; Jacob and de la Torre, 2009). Nonetheless, CK-666 treatment disrupted phagocytosis in confined and unconfined settings when compared to DMSO control (Fig. 2F; Movie EV4). Interestingly, confinement increased phagocytosis in the presence of CK-666 compared to performance in CK-treated unconfined conditions (Fig. 2F). These data suggest that there may

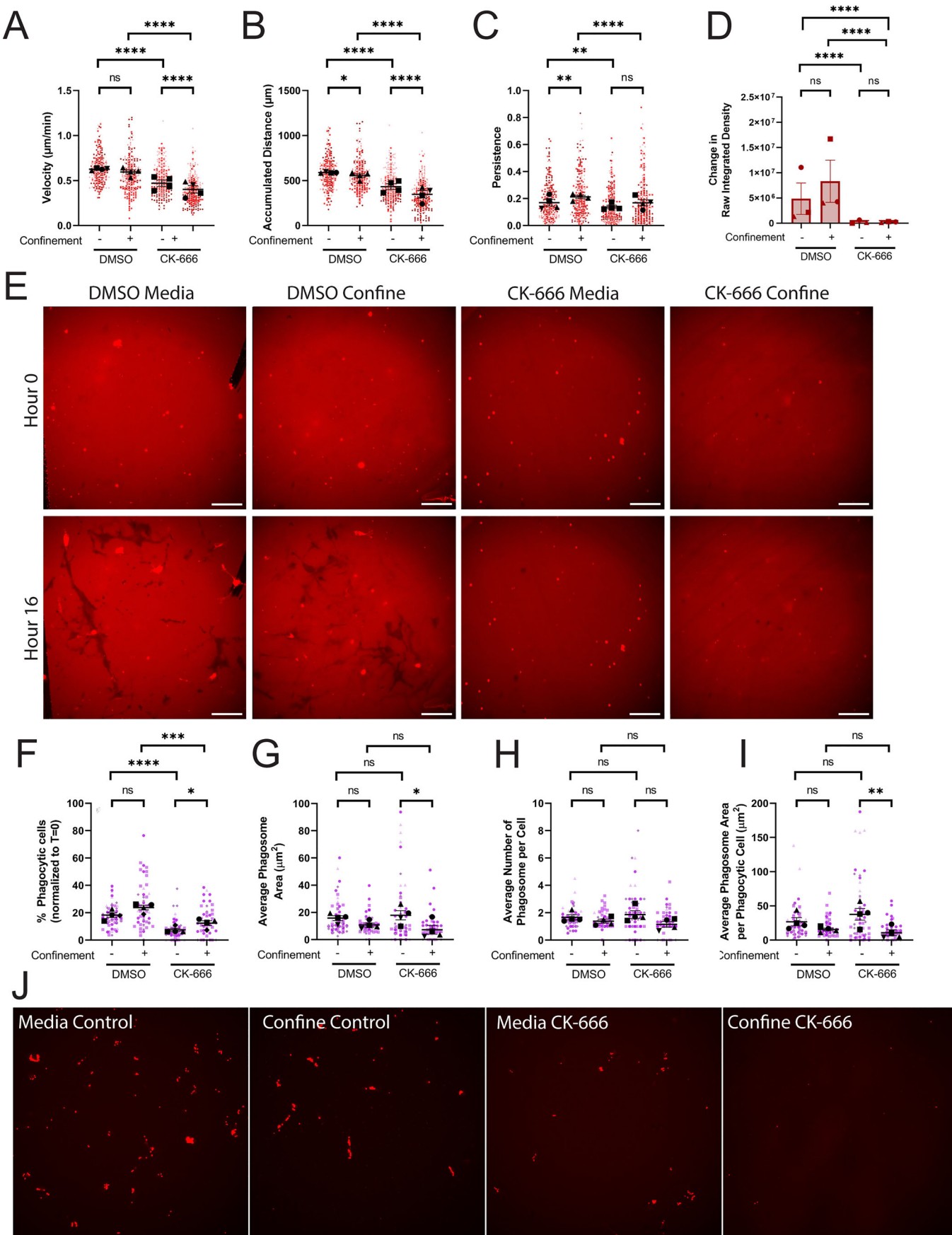

**Figure 2. The Arp2/3 complex is required for iC3b phagocytosis, but confinement rescue is context dependent.**

(A–E) Cells plated on 10 µg/mL AF-iC3b in each condition: Media control (vehicle – DMSO), Confined control (1% agarose) plus Vehicle, media control plus CK-666, confined (1% agarose) plus CK-666. (A) Velocity of cells migrating (µm/min). (B) The maximum accumulated distance (µm) that the cell traveled. (C) Persistence of the cell during the length of its track. (D) Quantification of amount of iC3b phagocytized off bottom of the dish, demonstrated in change in raw integrated density of fluorescence. (E) Example images of phagocytosis of AF-iC3b coating under each treatment condition after 16 h. Scale bar represents 100 µm. (F–J) Cells plated on 10 µg/mL FN in each condition and given pHrodo-iC3b beads: Media control (vehicle), Confined control (1% agarose) plus Vehicle, media control plus CK-666, Confined (1% agarose) plus CK-666. F) The percentage of fluorescent cells in a field of view, comparing unconfined and confined cells, normalized to T = 0 h. Any cell fluorescent at T = 0 was removed from further counting. (G) The average phagosome area per phagocytic cell (µm²). This is calculated by dividing the average size of all phagosomes in a field of view by the number of fluorescent cells counted (denoted phagocytic cells). (H) Average number of phagosomes per phagocytic cell. (I) Average phagosome size (µm²). This is the average size of phagosomes per field of view divided by the average number of phagosomes per field of view. (J) Example composite images of phase contrast and pHrodo-red merged images 8 h after pHrodo-iC3b beads were added to media wells (left) or confined agarose wells (right). The pHrodo-red signal from these merges has been isolated alongside the merge to emphasize internalized pHrodo-iC3b-bead signal. Scale bar represents 100 µm. For all graphs, black points demonstrate experiment means, colored points demonstrate individual cell values for each run. $N = 4$ experiments (biological replicates) for migration (A–E) or $N = 3$ experiments (biological replicates) for phagocytosis (F–J); $n = 50$ cells for each condition per experiment for migration (A–C), $n = 10$ fields of view per condition for coating phagocytosis (E), or $n = 15$ fields of view for each condition per experiment for phagocytosis (F–J). Statistical analysis was assessed using via Kruskal–Wallis test with Dunn multiple comparisons. Error bars represent SEM. Values for $p$ are as follows: (A) ns nonsignificant, ****$p < 0.0001$. B) *$p = 0.0463$, ****$p < 0.0001$. (C) **$p = 0.0015$ (DMSO media vs DMSO confined), **$p = 0.0025$ (DMSO media vs CK-666 media), ****$p < 0.0001$, ns nonsignificant. (D) ns nonsignificant, ****$p < 0.0001$. (F) ns nonsignificant, ****$p < 0.0001$, ***$p = 0.001$, *$p = 0.0126$. (G) ns nonsignificant, *$p = 0.0405$. (H) ns nonsignificant. (I) ns nonsignificant, **$p = 0.0081$. Source data are available online for this figure.

be additional factors beyond the Arp2/3 complex that influence phagocytosis in response to confinement. Phagocytic impairment did not result in a change in phagosome size or number between DMSO and CK-666-treated cells assessed in the same confinement context (Fig. 2G–J). Interaction of cells with surface-labeled AF-iC3b and bead-opsonized pHrodo-iC3b resulted in lowered cell migration and cell phagocytosis upon Arp2/3 inhibition. These results using surface-labeled AF-iC3b led us to next examine whether iC3b is a haptotactic cue.

## Arp2/3 complex is required for iC3b haptotaxis but not confinement-induced migration on iC3b

Haptotaxis is directed, persistent cell migration in response to an increasing concentration of a substrate-bound cue. Gradients of ECM proteins like fibronectin are well-characterized haptotactic cues, (Fortunato and Sunyer, 2022), but it is not clear whether complement proteins like iC3b stimulate a directional migratory response. To elucidate this possibility, we utilized the haptotaxis setup established previously in the lab to study fibronectin gradients (Stinson et al, 2025). In the present study, we generated a gradient of AF-iC3b with a 250 µg/mL source concentration in the central well (Fig. 3A). Wells were spaced 1.25 mm apart, and after letting the gradient establish for an hour, we seeded BV-2 cells into the outer wells and allowed them to migrate for 24 h. To ensure the gradient was present, we measured from the inner edge of one cell well across the center well to the inner edge of the second cell well at the start and end of each run (Fig. EV3A,B). These data confirm the consistency and stability of the iC3b gradient over time. Cell migration was assessed in what we previously established as "Zone 2", one field of view displaced from the edge of the left and right cell wells (Stinson et al, 2025). Notably, cells in each of the outer wells respond by moving under agarose with the gradient (+, toward the center well) or against it (−, away from the center well). Thus, it is possible to directly compare the two populations of cells moving in the + and – directions within the same experiment. TIRF imaging of the glass bottom of our dishes confirmed that iC3b is bound to the glass and forms a gradient under the agarose (Fig. EV3C). This finding is consistent with the finding that the mean fluorescence intensity of iC3b did not substantially decrease after a series of PBS washes (Fig. EV3D).

BV-2 cells responded haptotactically to the iC3b gradient. Example histograms of cell motility demonstrate more cells moving upward, along the gradient (right panel of Fig. 3B; Movie EV5) compared to cells moving away from the gradient, which had fewer cells moving as strongly along the gradient axis (left panel of Fig. 3B; Movie EV5). The forward migration index (FMIx) was significantly increased when cells moved with the gradient, compared to away from it (Fig. 3C), strongly suggesting a directional bias toward an iC3b gradient, which is indicative of a haptotactic response. Cell persistence mirrors the FMIx results, as cells moved with higher persistence toward the iC3b gradient (Fig. 3D). We also calculated effect sizes using Hedges' g and found values of 1.65 for FMI and 1.15 for persistence when comparing the (+) and (−) populations (Dataset EV1). Cells moving along the iC3b gradient (+) are slower compared to cells in the (−) orientation but showed no change in total distance (Fig. 3E,F). These analyses together suggest that cells moving toward the iC3b gradient do so with higher directionality than cells moving away from the gradient, which is consistent with haptotactic migration.

The Arp2/3 complex is a critical effector of fibronectin haptotactic sensing (Stinson et al, 2025; Wu et al, 2012). To assay this in BV-2 cells, we repeated our experiment with CK-666 polymerized into the agarose (Figs. 3G and Fig. EV3F,G). When exposed to CK-666, BV-2 cells moving along the iC3b gradient had more highly spread migration tracks compared to untreated cells moving under the same conditions (compare 'toward gradient' populations in Figs. 3H and 3B), similar to cells moving against the gradient in both datasets. FMIx quantification demonstrates this further, as there is no statistical difference between cells moving toward (+) or away (−) from the gradient when CK-666 is present (Fig. 3I; Movie EV6). While untreated cells increase migratory persistence and decrease velocity on an iC3b gradient (+population) (Fig. 3D), both measures are similar in (+) and (−) populations when CK-666 is present (Fig. 3J,K). While Arp2/3 is required for directional migration under these conditions, Arp2/3-deficient macrophages (Stinson et al, 2025), microglia (the current

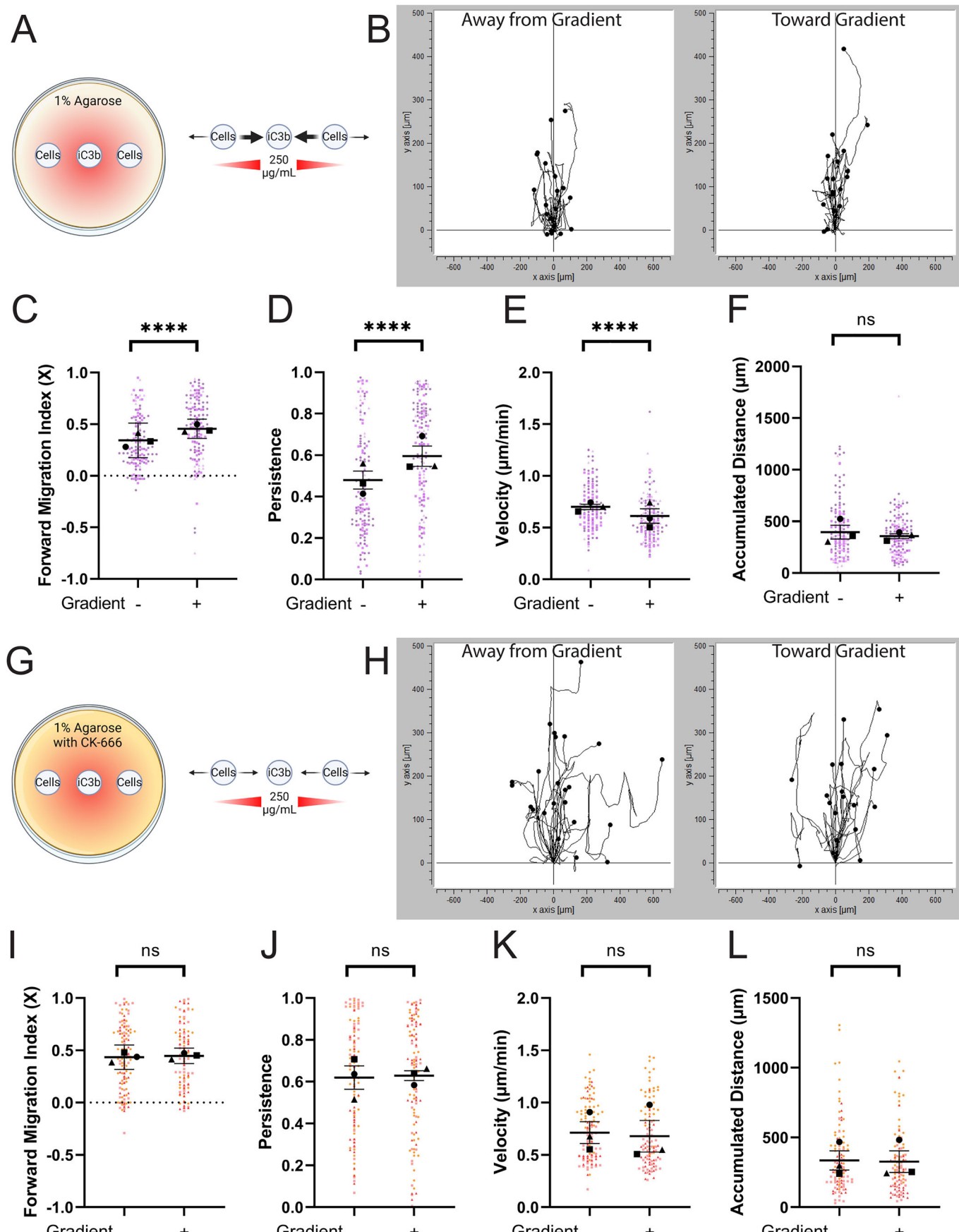

**Figure 3. The Arp2/3 complex is required for the haptotactic response to iC3b.**

(A) Schematic depicting haptotaxis dish layout. Wells are placed 1.25 mm apart. 250 µg/mL AF-iC3b is placed in the center well and allowed to spread for 1 h before washing. Cells are added to outer wells and then allowed to move under the agarose for 24 h either towards the center well (toward gradient, denoted (+)) or away from the center well (away from gradient, denoted (−)). Strength of movement is noted in arrow thickness. (B) Example histograms of cell tracks demonstrating the number of cells traveling in each direction. (C) Forward migration index in the X direction, measuring cell tracks in a binary system moving toward or away from the AF-iC3b haptotactic cue. (D) Persistence of the cell during the length of its track. (E) Velocity of cells migrating (µm/min). (F) The maximum accumulated distance (µm) that the cell traveled. (G) Schematic depicting haptotaxis apparatus with the addition of 125 µM CK-666 to agarose. Strength of cellular response to gradient cue is noted in arrow thickness. (H–L) same type of graphs as in (B–F), noting now the addition of CK-666 to impact cell motility. For all graphs, black points demonstrate experiment means, colored points demonstrate individual cell values for each run. $N = 3$ experiments (biological replicates) for each graph; $n = 50$ cells for each direction per experiment for migration. Statistical analysis was assessed using the Mann–Whitney test. Error Bars represent SEM in all graphs except for FMIx, where Error bars represent a 95% confidence interval. Values for $p$ are as follows: (C) ****$p < 0.0001$. (D) ****$p < 0.0001$. (E) ****$p < 0.0001$. (F) ns nonsignificant. (I) ns nonsignificant. (J) ns nonsignificant. (K) ns nonsignificant. (L) ns nonsignificant. Source data are available online for this figure.

study), and dendritic cells (Vargas et al, 2016) are all capable of moving well under agarose. As with the untreated cells, CK-666 had no effect on distance traveled during migration (Fig. 3L). Hedges' g values suggest that the effect size was low when comparing (+) and (−) CK-666-treated populations for both FMI (0.25) and persistence values (0.099). These analyses together suggest that Arp2/3 complex disruption causes cells to no longer prefer the (+) condition over the (−) condition, which is consistent with loss of haptosensing.

We repeated this experiment with the small molecule CK-689 polymerized into the agarose (Fig. EV3E). CK-689 is an inactive control compound for CK-666, which lacks the ability to inhibit the Arp2/3 complex and thus acts as a negative control for our CK-666 experiments (Fokin et al, 2022; Aloisio and Barber, 2022). Cells treated with CK-689 moved similarly to untreated cells on iC3b gradients (Figs. 3C–E and EV3E), further supporting the case for the CK-666 haptotaxis phenotype. These data together suggest that iC3b is a haptotactic cue, and that Arp2/3-generated lamellipodial protrusions are necessary for microglia to detect iC3b gradients.

## The Arp2/3 complex maintains microglial morphology in situ

We then translated our in vitro results into a system with more in vivo relevance: live imaging of microglia in situ in a hippocampal slice model. Utilizing CX3CR1$^{+/GFP}$ mice (Jung et al, 2000), we created 400 µm coronal slices of mouse brains from mice between 35 and 50 days old, of equal sex representation, and imaged in the CA1 region of the hippocampus. To examine the effects of Arp2/3 complex inhibition, slices were incubated in either 200 µM CK-666 or DMSO vehicle control prior to imaging. The 200 µM CK-666 dose was chosen based on previous literature utilizing CK-666 in hippocampal slices (Drew et al, 2020).

We started by examining microglial surveillance. Individual cells were imaged every minute for 10 min to examine dynamic movement under our different treatment groups: no treatment (negative control), DMSO (vehicle control), or 200 µM CK-666 (images and skeletons, Fig. 4A). First, we measured the morphological changes in cells when treated with either DMSO or CK-666 at the first time point in our surveillance videos. CK-666 treatment significantly decreased the number of branches, junction points, and process tips compared to both DMSO-treated and non-treated cells (Fig. 4B). CK-666 also increased the proportion of the cell soma relative to the total cell area (Fig. 4C), pointing towards a compensatory increase in soma size when processes are no longer

being formed efficiently. These data indicate that the Arp2/3 complex plays a major role in maintaining microglial morphology in situ. There was no statistically significant difference between the DMSO and non-treated cells.

We then examined changes in cell dynamics over time. Overall, we saw similar results in our non-treated and our DMSO-treated cells, with CK-666 cells demonstrating significantly different values in many morphology measurements (Fig. 4D). As with our single timepoint analysis, CK-666 treatment caused microglia to have fewer branches, fewer junctions, and fewer tips along their existing branches over the entire time course (Fig. 4D). The overall number of processes are limited as well in CK-666-treated microglia, reflected by a lower ramification index (overall complexity of the microglial skeleton) and a smaller convex hull measurement (spanned area of the cells) as the cells are taking less area within their 3D environment compared to untreated and DMSO-treated counterparts (Fig. 4D).

We also examined whether surveillance-like behaviors were impacted by Arp2/3 disruption. Indeed, CK-666 treatment impairs protrusion (extensions) and retraction compared to controls (Fig. 4E, quantified in 4F). Interestingly, DMSO treatment increased protrusion (extensions) and retraction compared to untreated counterparts, suggesting that vehicle treatment elicits a somewhat more dynamic surveillance behavior in these cells (Fig. 4F). Given the profound contribution of Arp2/3 to maintaining microglial morphology and supporting surveillance-related cell protrusion, we wanted to next evaluate how these alterations in cellular structure and behavior would impact microenvironmental sensing of chemotactic and haptotactic cues.

## Arp2/3 complex is required for iC3b sensing but not ATP-induced chemotaxis

The Arp2/3 complex is necessary for macrophage haptotaxis but not chemotaxis (27). However, the differential contribution of Arp2/3 to specific extracellular sensing pathways has not been as robustly studied in vivo in mice. Our hippocampal slice culture presents an ideal opportunity to evaluate this question using endogenous microglia in a physiologically relevant setting in situ. We decided to use ATP for in situ chemotaxis, due to its established role as a microglial chemotactic cue (33–35). We developed a system to inject vehicle (tissue bath solution) or 1 mM ATP into DMSO- or CK-666-treated brain slices using a micropipette. We imaged brain slices for 1 h at 2-min intervals, with ATP injection at 6 min after establishing a baseline for three images (reflected in

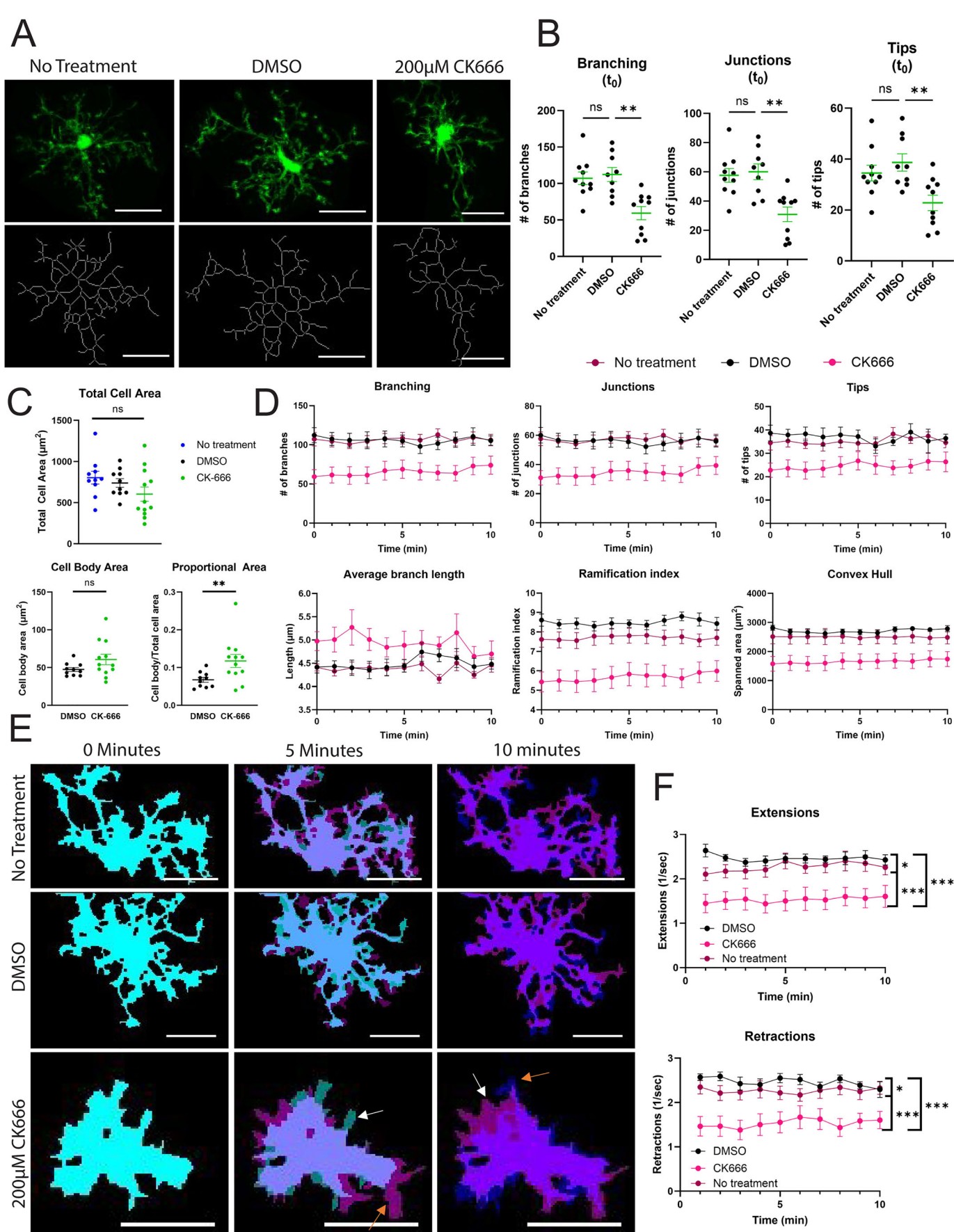

**Figure 4. Disruption of the Arp2/3 complex in situ alters microglial morphology.**

(A) Maximum intensity z-projections of original images (top) with skeletons generated using MotiQ 2D Analyzer (bottom) at first imaging timepoint after drug washout. Scale bar represents 20 µm. The CK-666-treated image is cropped to isolate and highlight one cell from a group of three. The uncropped image is provided as accompanying source data. (B) Quantification of branching, junctions, and tips at first imaging timepoint ($n = 9$–10 cells). Statistical analysis performed using one-way ANOVA (**$p$ value $= 0.0010$ (branching), **$p$-value $= 0.0011$ (junctions), **$p$ value $= 0.0047$ (tips)), lines and error bars represent mean ± SEM. (C) Measurements of total cell area (top), cell body (bottom left), and the proportion of the cell body to the total cell area (bottom right). The error bars represent mean ± SEM. For statistical tests, an ordinary one-way ANOVA was used to compare groups in the total cell area graph ($n = 10$–12 cells). Data were normally distributed according to all normality tests and the appearance of the QQ plot. Variances did not significantly differ between groups. Based on normal distribution (confirmed with QQ plot visualization) but significantly different variances, a Welch's $T$-test was performed to compare area between DMSO and CK-666 treatment for both cell body and cell proportions graphs ($n = 10$–12 cells). (ns notsignificant, **$p = 0.0164$). (D) Minute-to-minute morphological features after drug washout (mean ± SEM) ($n = 9$–10). (E) Binarized image with "mean" algorithm and closed function. Colorized masks represent independent timepoints (0 min = cyan, 5 min = magenta, 10 min = blue). Non-overlapping regions represent areas of extensions and retractions. White arrows pointing examples of retraction; orange arrows pointing examples of extension. Scale bar represents 20 µm. (F) MotiQ automated quantification of extensions and retractions (mean ± SEM) ($n = 9$–10 cells). Statistical analysis performed using an ordinary 2-way ANOVA with Tukey's multiple comparisons test of treatments. Main effect of drug treatment ($F(2, 258) = 74.71$, ***$p < 0.0001$) with no time-dependent effect or interaction. Between group comparisons were significant (Extensions: ***$p < 0.0001$, *$p = 0.0427$; Retractions: ***$p < 0.0001$, *$p = 0.0415$). Source data are available online for this figure.

"6 min" timestamp indicator in the leftmost images in Fig. 5A), and continued imaging for 52 min. After the ATP injection, a ring of microglial processes extended toward the pipette end until the pipette itself was blocked (see circular ROI in Fig. 5A; Movie EV7), resulting in two occurrences: (1) one or multiple processes climbed the inside of the pipette to further respond to the ATP and (2) the external processes that had been responding stopped, as there was no longer a chemotactic source of ATP upon blockage of the pipette. DMSO- and CK-666-treated cells responded similarly to ATP. Microglia did not respond to the injection of tissue bath solution (Fig. 5B, left), as 54 min elapsed without a response. ATP injection elicited a rapid response that was unaffected by Arp2/3 inhibition (Fig. 5B, right). Likewise, ATP elicited process elongation equally from DMSO- and CK-666-treated microglia, while bath solution failed to elicit process responses (Fig. 5C). Finally, both DMSO- and CK-666-treated microglial processes were equally capable of extending up the ATP-containing pipette until they clogged it (Fig. 5D). When ATP-treated mice were separated by sex, we only noted a faster response time in our male mice for CK-666-treated microglia (Fig. EV4A), while none of the other measurements were significant (Fig. EV4A–C). Next, we quantified process motility via manual tracking and determined that velocity, accumulated distance and persistence in response to ATP were all unaffected by Arp2/3 impairment (Fig. 5E–G). When ATP-treated mice were separated by sex, no significant relationships were revealed in any of these migration measures (Fig. EV4D–F). In total, these data suggest that microglial process generation and chemotactic sensing in vivo is not likely to be impaired by loss of Arp2/3 function.

Next, we adapted the micropipette injection protocol to incorporate 2-micron pHrodo-iC3b beads to test whether microglia demonstrate an Arp2/3-dependent haptotactic response in situ. In these experiments, hippocampal slices were imaged for 2 h with 2-min intervals. pHrodo-iC3b beads were injected at the 6-min mark, similarly to ATP injection (Fig. 6A; Movie EV8). Remarkably, DMSO-treated microglia stably interact with iC3b beads and consolidate them over time (Fig. 6A, as indicated by the yellow arrow, and Movie EV8). On the other hand, microglia treated with CK-666 still extend processes toward the iC3b beads, but fail to stably interact with them, as the beads remain within the region of interest for the duration of the time course (Fig. 6A, bottom). At the end of the 2-h time course, many beads could be found associated with DMSO-treated microglia, while the interaction of

CK-666-treated microglia with beads remained suppressed (Fig. 6B; Movies EV9 and 10). As mentioned previously, this difference does not seem to be due to an inability of Arp2/3-deficient microglia to make protrusions in response to iC3b, as their process length remained similar to DMSO-treated counterparts (Fig. 6C). Instead, this phenotype seems to stem from an inability of Arp2/3-deficinet microglia to make stable, long-lasting contact with iC3b (Fig. 6D). We did not observe any sex differences with respect to process length or bead interaction (Fig. EV4G,H). The relationship between DMSO and CK-666 treatment in Fig. EV4H is similar in male and female samples, suggesting that this effect is not dependent upon sex. These data were consistent with defective in situ haptotaxis when Arp2/3 is impaired. We next used manual tracking to determine the migratory character of these two populations. CK-666 treatment significantly increased velocity and accumulated distance compared to DMSO control when microglia responded to iC3b (Fig. 6E,F). Conversely, CK-666 decreased migratory persistence in this context compared to control (Fig. 6G). Again, we demonstrated similar relationships in samples stratified by sex, suggesting that this effect is not dependent upon sex (Fig. EV4I–K). These tracking data suggest that Arp2/3-deficient microglia generate highly dynamic, faster-moving protrusions that randomly contact and immediately release iC3b-labeled targets. This is in direct contrast to Arp2/3-deficient microglia responding to ATP, which demonstrated process dynamics statistically indistinct from control cells. Local iC3b gradients could help recruit microglia to areas where they are required for spine pruning without requiring a large response from the regional population. Together, our data from an in vivo-like environment suggest that Arp2/3 complex-dependent haptosensing could be relevant for microglial clearance of iC3b-labeled targets in the CNS.

## Discussion

We set out to examine the Arp2/3 complex's contribution to microglial function, and to determine the nature of iC3b sensing in vitro and in situ. BV-2 cells move less on iC3b than on fibronectin, likely due to their propensity to phagocytose iC3b from the bottom of the dish in an Arp2/3-dependent fashion. With iC3b-opsonized beads, phagocytosis increased under confinement, again in an Arp2/3 complex-dependent fashion. Likewise, when these cells were tasked with moving under confinement on iC3b in the

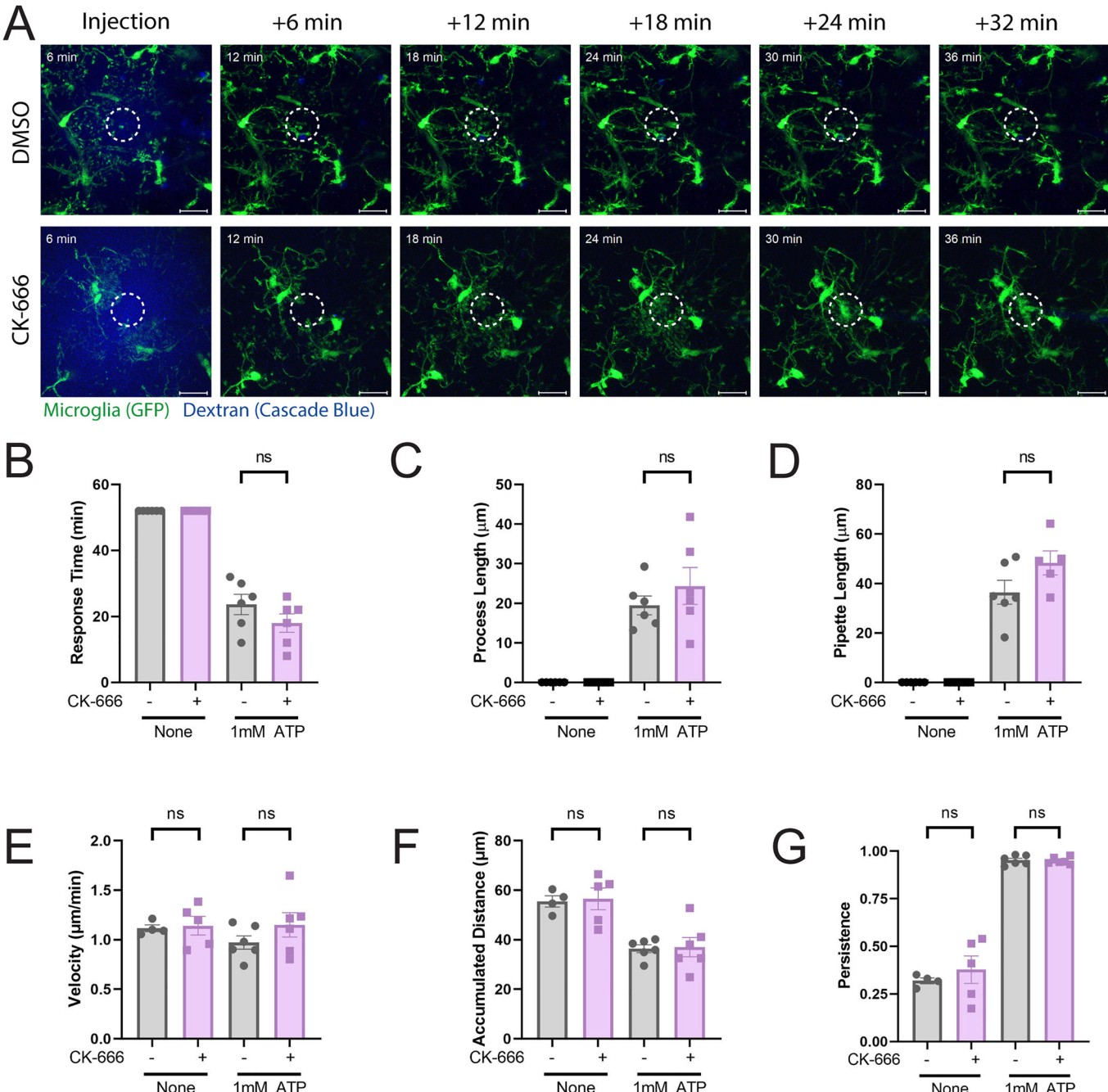

**Figure 5. The Arp2/3 complex is not required for cellular response to ATP-dependent chemotaxis in situ.**

(A) Example time course of chemotactic response to 1 mM ATP injection with vehicle (DMSO) or CK-666 treatment. The circle represents the region of interest, denoting the injection tip. Scale bars represent 20 μm. These images have been cropped from the full-size image to accentuate the ROI that denotes the injection area. The uncropped images have been included as source data with this publication. (B) Measured response time from cells receiving a chemotactic cue of 1 mM ATP to the processes blocking the pipette, in minutes. (C) Averaged process length (μm) from cells responding to chemotactic cue. (D) Length (in μm) of micropipette that a process has visibly extended into by the end of the 1 h video. (E) Velocity of processes moving post-injection (μm/min). (F) The maximum accumulated distance (μm) that the process traveled. (G) Persistence of the process during the length of its track. $N = 6$ experiments for each graph (3 male; 3 female biological replicates). Statistical analysis was assessed using the Mann–Whitney test. Error bars represent SEM in all graphs. Values for p are as follows: (B) ns nonsignificant. (C) ns nonsignificant. (D) ns nonsignificant. (E) ns nonsignificant. (F) ns nonsignificant. (G) ns nonsignificant. Source data are available online for this figure.

 

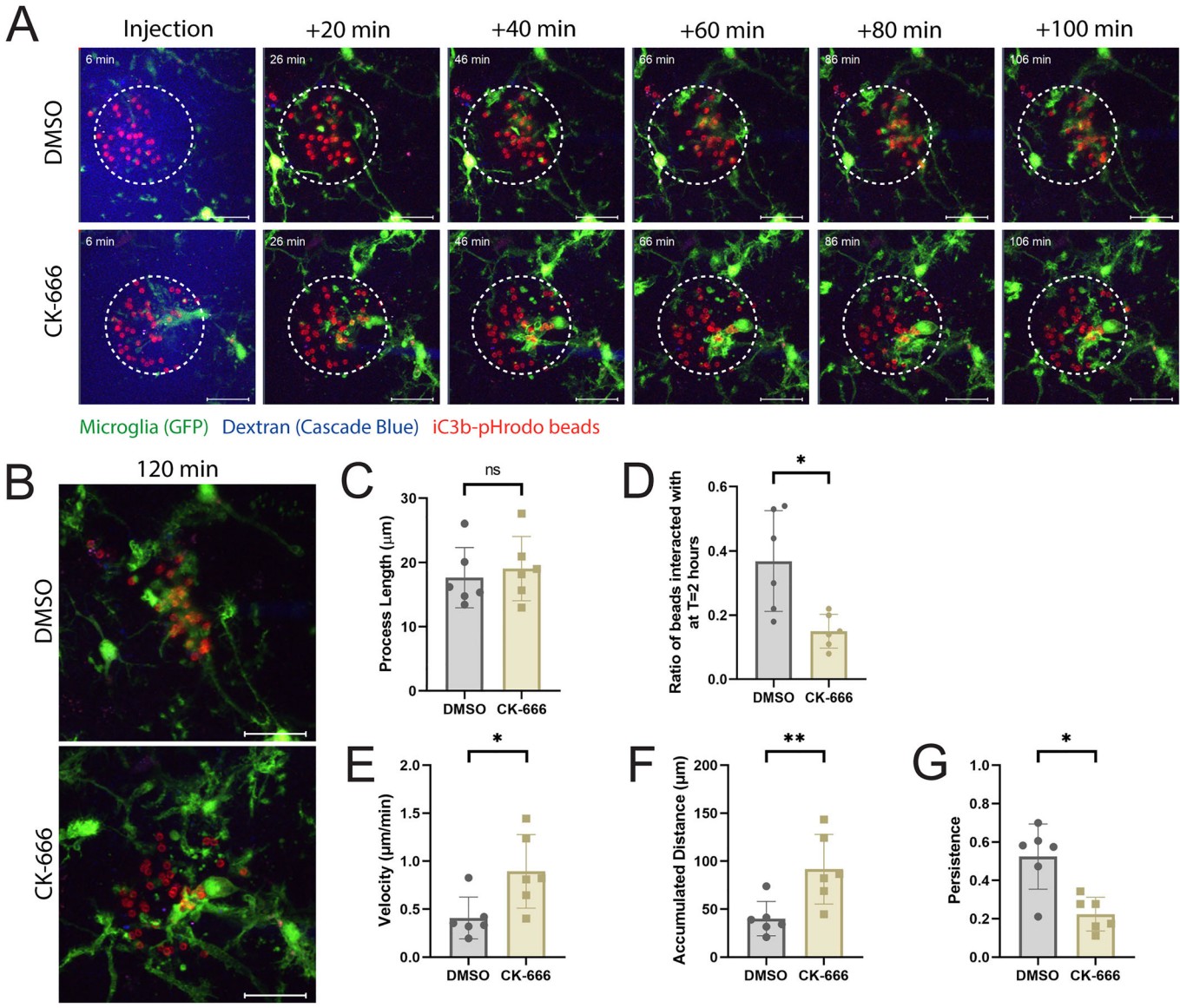

**Microglia (GFP)  Dextran (Cascade Blue)  iC3b-pHrodo beads**

**Figure 6. The Arp2/3 complex is required for cellular response to iC3b in situ.**

(A) Example time course of cellular response to pHrodo-iC3b-opsonized bead injection with vehicle (DMSO) or CK-666 treatment. The circle represents the region of interest, demonstrating the starting boundary of bead injection. These images have been cropped from the full-size image to accentuate the ROI around the beads. The uncropped images have been included as source data with this publication. (B) Example endpoint images of 2-h phagocytosis assay. Scale bar represents 20 μm. (C) Averaged process length (μm) from cells responding to pHrodo-iC3b beads cue. (D) Ratio of beads interacted with vs not interacted with at the end of the 2-h assay. (E) Velocity of processes moving post-injection (μm/min). (F) The maximum accumulated distance (μm) that the process traveled. (G) Persistence of the process during the length of its track. $N = 6$ experiments for each graph (3 male; 3 female biological replicates). Statistical analysis was assessed using the Mann–Whitney test. Error Bars represent SEM in all graphs. Values for p are as follows: (C) ns nonsignificant. (D) *$p = 0.0108$. (E) *$p = 0.0411$. (F) **$p = 0.0087$. (G) *$p = 0.0152$. Source data are available online for this figure.

presence of CK-666, we see a decrease in migratory abilities, including loss of haptotaxis and a decrease in phagocytosis that is partially rescued by confinement. Finally, we translated these findings into in situ studies of microglia in brain slices. The Arp2/3 complex is responsible for baseline surveillance and maintaining ramified microglia morphology in this context, based on our CK-666 experiments and experiments by Safaiyan and co-authors (Safaiyan et al, 2026) investigating *Arpc4−/−* microglia in situ. While the chemotactic response of microglia processes to ATP was

maintained in CK-666-treated slices, CK-666-treated microglia could not extend stable processes that maintain contact with iC3b-labeled beads in situ when compared to vehicle-treated microglia. Together, this body of work establishes that Arp2/3 complex-dependent haptosensing could be an important feature of the microglial response to iC3b.

Several isoforms exist for specific subunits of the Arp2/3 complex. Their differential incorporation into the complex gives rise to versions of the Arp2/3 complex with different activities. While some cell types

express only one isoform or the other, mouse microglia are expected to express several of the isoforms including *Arpc1a*, *Arpc1b*, *Arpc5*, and *Arpc5L* (data from brainrnaseq.org, (Bennett et al, 2016)). Abella and colleagues demonstrated the striking finding that Arpc1B- and Arpc5L-containing complexes promoted actin assembly better than those that had Arpc1A or Arpc5 (Abella et al, 2016). Furthermore, cortactin was shown to stabilize Arpc1B- and Arpc5L-containing filaments more efficiently, thereby shielding these complexes from coronin-induced debranching (Abella et al, 2016). In particular, Arpc1B seems to be critical for immune cell function, as its loss in T cells and platelets leads to immunodeficiencies in human patients (Kahr et al, 2017; Somech et al, 2017; Brigida et al, 2018). Its loss in B cells and macrophages also leads to defects in lamellipodial and podosome function (Leung et al, 2021). These data suggest that the different forms of the Arp2/3 complex do not perfectly compensate for one another. It has also been reported that CK-666 possesses different efficacy depending on whether Arpc1A or Arpc1B is present, with Arpc1B complexes being more resistant to the drug in the context of canonical branched actin formation via Type 1 nucleation promoting factors (NPFs) (Cao et al, 2024). However, the authors did note that production of linear filaments via SPIN90-mediated activation of the complex (Wagner et al, 2013) was blocked well by CK-666 in either case (Cao et al, 2024). It has also been proposed that an alternative complex composed of Arp2, Arp3, Arpc2 ± Arpc3 binds to vinculin, and may mutually antagonize Arpc1-containing complexes, which may have implications for focal adhesion function (Chorev et al, 2014). One possibility based on these previous studies is that Arpc1A-containing complexes are important for microglial function, given that we see significant reductions in iC3b phagocytosis, iC3b uptake from glass, and impaired haptotactic sensing in response to CK-666. Another possibility is that SPIN90-dependent linear actin produced by the Arp2/3 complex is uniquely important for these processes. It is also possible that microglia simply respond differently than bone marrow-derived macrophages to CK-666. However, the effect of CK-666 on haptotaxis suggests that it could disrupt proper focal adhesion function, perhaps by impairing Arpc1A complexes and/or altering the activity of vinculin-binding "hybrid Arp2/3 complexes".

Though haptotaxis has been interrogated at the molecular level in vitro, much less is known about how haptosensing contributes to physiological functions. Leukocyte haptotaxis is perhaps better contextualized in vivo than for other cell types. One striking example is dendritic cells' ability to haptotax along a gradient of immobilized chemokine CCL21, with compelling evidence pointing to in vivo CCL21 gradients (Schwarz et al, 2017). Within the CNS, axonal growth cones of newly developing neurons in both the brain and spinal cord follow gradients of both chemoattractant and haptoattractant cues, with new studies pointing towards netrin and sonic hedgehog as haptotactic cues for neurons (Wu et al, 2019; Qiu et al, 2024). Recent studies have found that calvarial osteoblast progenitors haptotax along the embryonic cranial mesenchyme (Liu et al, 2024). Conversely, disruption of haptosening can be detrimental. Tumor cell invasion has been shown to downregulate actin cytoskeleton components responsible for haptotactic sensing to enable this migration (Chan et al, 2014; King et al, 2016). Our work contributes to this burgeoning area by demonstrating that iC3b elicits a haptotactic response, and that disrupting microglial haptosensing in the context of the CNS microenvironment impairs the response to complement-opsonized ligands. The role of the Arp2/3 complex in integrin-dependent phagocytosis and motility

has been established in vitro (Rotty et al, 2017). In addition, iC3b phagocytosis proceeds via a molecular clutch system highly similar to the mechanism found at focal adhesions (Jaumouillé et al, 2019). Conversely, studies in multiple cell types confirm that Arp2/3 is dispensable for chemotaxis (Wu et al, 2012; Rotty et al, 2017; Dimchev et al, 2021; Vargas et al, 2016; Paterson and Lämmermann, 2022). However, this dichotomy has not been thoroughly explored in mammalian systems in vivo. In the present study, we demonstrated that Arp2/3 complex-deficient microglia in their in situ environment respond dramatically to a chemotactic ligand while failing to interact with local iC3b cues. Safaiyan *et al* found a higher percentage of *Arpc4*−/− microglia with internalized myelin compared to control, suggesting an overexuberant phagocytic response that is Arp2/3-independent (Safaiyan et al, 2026). Together, these studies suggest that microglia can integrate cues from their environment using distinct sensing pathways, and that pathways with similar biological functions (e.g., iC3b versus myelin phagocytosis) may operate via distinct molecular pathways. Thus, cytoskeletal responses to the microenvironment may be much more specific than previously suspected.

The complement cascade possesses qualities that suggest the possibility of in vivo haptotactic gradients. First, it is locally activated by an initiating stimulus (Ricklin et al, 2010). This local response leads to the production of activated complement proteins, including C3b (Ricklin et al, 2010). C3b is highly reactive, so it is unlikely to diffuse long distances like chemotactic cues (Sim et al, 1981). Instead, C3b rapidly binds covalently to hydroxyl or amine groups (Ricklin et al, 2016) to label many targets, including pathogen surfaces, immune complexes, and apoptotic cells (Mastellos et al, 2024). In addition, C3 has been detected at amyloid plaques (Ishii & Haga, 1984; Veerhuis et al, 1995; Torvell et al, 2021) and is associated with extracellular matrix proteins (Balduit et al, 2025). After deposition, C3b is then converted to iC3b, which remains immobilized (Ricklin et al, 2010). The concentration of both C3b and iC3b would be highest closer to the site of activation and would likely decrease proportionally to the distance from the initiating site. Finally, microglia possess several surface receptors, like CR3 (αMβ2 integrin) (Zhang et al, 2014; Stephan et al, 2012), that can detect the outlying region of the gradient and "read" it to move toward the site of highest complement activity. A haptotactic complement gradient could be relevant to microglia interaction with amyloid plaques, as mentioned above. However, complement proteins also interact with ECM components like heparan sulfate (Yamaguchi, 2001), hyaluronan (Wilson et al, 2020), chondroitin sulfate (Miller and Hsieh-Wilson, 2015), and aggrecan (Rowlands et al, 2018) that surround dendrites and/or spines (Balduit et al, 2025). These ECM-stabilized gradients of complement proteins could be organized such that microglia use them to orient their processes toward dendritic spines that need to be pruned.

Two different models of synaptic pruning have been proposed: direct trogocytosis via microglial interaction with dendritic spines, or neural shedding, wherein the spine detaches from the neuron and microglia phagocytose these shed spines within the ECM (Pereira-Iglesias et al, 2025). During trogocytosis, phagocytes pinch off pieces of a cell rather than fully ingesting them (Miyake and Karasuyama, 2021). Studies on synaptic pruning support the idea that microglial clearance of the presynaptic bulb at the axon terminal occurs via trogocytosis, both in hippocampal slices in situ (Lim and Ruthazer, 2021) and in amphibians in vivo (Weinhard et al, 2018). While previous studies have called postsynaptic clearance complement-mediated

phagocytosis (Tremblay et al, 2010; Schafer et al, 2012), the molecular pathway of trogocytosis may be quite similar. For example, the Arp2/3 complex creates part of the cell tunnel involved in forming the trogocytic cup (Watanabe et al, 2020), and is also required for iC3b phagocytosis. iC3b surface coating assays allow for the visualization of iC3b uptake via a process potentially similar to trogocytosis, which may facilitate modeling synaptic pruning in vitro. The other alternative, neural shedding, instead relies on neuronal autonomous regulation and microglia acting as scavengers for the iC3b label (Pereira-Iglesias et al, 2025; Chang et al, 2017; Blackman et al, 2012; Henson et al, 2017; Yuste and Bonhoeffer, 2001). Reminiscent of a gradient, shed dendritic spines could diffuse into the ECM, with the associated iC3b acting as a haptotactic ligand. This process would still drive microglial interaction, inflammatory responses mediated by iC3b, and engulfment of synaptic ends (Vandendriessche et al, 2021). In each of these scenarios, immobilized iC3b could serve as a local cue to either recruit microglia to dendritic spines for trogocytosis or to directly internalize shed dendritic spines. Future studies will further clarify how iC3b-dependent haptosensing contributes to microglial synaptic pruning.

Aberrant synaptic pruning and chronic neuroinflammation have been implicated in numerous neurodegenerative diseases, including Alzheimer's and Parkinson's disease (Kinney et al, 2018; Zhang et al, 2023; Gomez-Arboledas et al, 2021; Stephan et al, 2012; Werneburg et al, 2020; Yoshiyama et al, 2007; Maezawa and Jin, 2010; Wilton et al, 2023). Evidence points toward activated microglia phagocytosing dendritic spines prematurely during neurodegeneration, thereby contributing to cognitive decline via hastened synapse loss (Rosales-Corral et al, 2012). Multiple neurological disease mouse models have found that ablation of C1q (a complement component upstream of C3), C3, or CR3 improved learning and memory tasks (Warwick et al, 2021; Bourel et al, 2021; Wei et al, 2021; Silverman et al, 2019; Borucki et al, 2020). Microglial ablation has also improved learning and memory tasks in neurological disease model mice as well, suggesting that this glial cell population is potentially a major pathological driver (Willis et al, 2020; Henry et al, 2020). These studies suggest a need to mitigate hyperactive synaptic pruning by microglia in neurodegenerative diseases. The Arp2/3 complex might serve as a proof of concept for this idea. Inhibition of the Arp2/3 complex selectively targets haptosensing of iC3b but preserves microglial chemotaxis and process generation in response to ATP. Safaiyan et al show that $Arpc4^{-/-}$ microglia do not respond to laser-induced wounds, suggesting a defect in chemotactic sensing upon Arp2/3 disruption (Safaiyan et al, 2026). However, this is likely due to P2RY12 loss in these cells, which is likely intact in our acute CK-666 treatment model. Prior in vivo research involving the Arp2/3 complex in the CNS centers around its important role in dendritic spine formation in neurons (Kim et al, 2013a). Considering these studies, utilizing small-molecule approaches to disrupt Arp2/3 itself will likely have side effects beyond targeting synaptic pruning. This is vividly demonstrated by Arpc4 knockout in microglia, which disrupts myelin integrity, downregulates microglial TGF-β signaling in microglia, and shifts them to a more reactive phenotype (Safaiyan et al, 2026). Taking this into account, targeting iC3b haptosensing in microglia as specifically as possible upstream of Arp2/3 may effectively pause synaptic pruning. This intervention might preserve synapses and ameliorate symptoms of neurodegeneration. Preclinical studies on microglial haptosensing will need to be conducted in mouse models to fully understand that pathway's therapeutic potential and role in disease.

# Methods

### Reagents and tools table

| Reagent/resource | Reference or source | Identifier or catalog number |
|---|---|---|
| **Experimental models** | | |
| B6.129P2(Cg)-Cx3cr1tm1Litt/J | Jackson Laboratory | RRID:MGI:3580076 |
| C57BL/6 | Jackson Laboratory | RRID:MGI:2159769 |
| BV-2 cell line | Dr. Barrington Burnett | RRID:CVCL_0182 |
| **Recombinant DNA** | | |
| N/A | | |
| **Antibodies** | | |
| Normal Rat IgG | Invitrogen | Catalog #31933 |
| Integrin αM (Rat) | Abcam | Catalog #ab8878 |
| Integrin β2 (Rat) | NovusBio | Catalog #NBP1-41272 |
| **Oligonucleotides and other sequence-based reagents** | | |
| N/A | | |
| **Chemicals, enzymes and other reagents** | | |
| iC3b | CompTech | Catalog #A115 |
| CK-666 | Abcam | Catalog #Ab141231 |
| Anhydrous DMSO | Thermo Fisher | Catalog #D12345 |
| CK-689 | Sigma-Aldrich | Catalog #182517-25MG |
| Cascade Blue labeled 3000 MW Dextran | Thermo Fisher | Catalog #D7132 |
| NaCl | Sigma-Aldrich | Catalog #S9888-2.5KG |
| KCl | Sigma-Aldrich | Catalog #P3911-500G |
| $NaH_2PO_4$ | Sigma-Aldrich | Catalog # 71496-250 G |
| $MgSO_4 \cdot 7H_2O$ | Millipore Sigma | Catalog # 230391-25 G |
| $MgCl \cdot 6H_2O$ | Sigma-Aldrich | Catalog #M9272-100G |
| $NaHCO_3$ | Sigma-Aldrich | Catalog # S6014-500G |
| $CaCl_2 \cdot 2H_2O$ | Sigma-Aldrich | Catalog # 223506-500 G |
| D-glucose | Sigma-Aldrich | Catalog # G7021-100G |
| Sucrose | Sigma-Aldrich | Catalog # S0389-500G |
| Human serum fibronectin | Thermo Fisher | Catalog # 33016015 |
| low gelling temp, agarose 25 g | Sigma-Aldrich | Catalog # A9045-25G |
| Bovine serum albumin (BSA) | Sigma-Aldrich | Catalog # A9418-100G |

| Reagent/resource | Reference or source | Identifier or catalog number |
|---|---|---|
| Hanks Balanced Salt Solution (Powder) | Sigma-Aldrich | Catalog # H1387-10X1L |
| ATP | Sigma-Aldrich | Catalog # A2383-5G |
| DMEM, high glucose, no glutamine, no phenol red | Thermo Fisher | Catalog # 31053-036 |
| Fetal bovine serum (FBS) | Sigma-Aldrich | Catalog # 12306C-500ML |
| GlutaMAX™ Supplement | Thermo Fisher | Catalog # 35050061 |
| Antibiotic/Antimycotic | Thermo Fisher | Catalog # 15240062 |
| Trypsin-EDTA Solution 0.05% | Caisson Labs | Catalog # TRL02 |
| Corning™ Cell Culture Phosphate Buffered Saline (1X) | Fisher Scientific | Catalog # MT21040CV |
| Cell Culture Grade Water | Corning | Catalog # 25-055-CV |
| **Software** | | |
| Olympus CellSens acquisition and analysis Software | http://www.olympus-lifescience.com/en/software/cellsens/ | RRID:SCR_014551 |
| Fiji ImageJ2 | http://fiji.sc/ | RRID:SCR_002285 |
| ImageJ Manual Tracking plugin | https://imagej.net/ij/plugins/track/track.html | Included in Fiji ImageJ2 |
| ImageJ Chemotaxis Tool plugin and stand-alone Ibidi Chemotaxis and Migration Tool | https://ibidi.com/chemotaxis-analysis/171-chemotaxis-and-migration-tool.html | Version 1.01 (ImageJ plugin) for quantification of manual tracking Version 2.0 (stand-alone) for histogram creation |
| ImageJ Cell Counter plugin | https://imagej.net/ij/plugins/cell-counter.html | RRID:SCR_025376 Included in Fiji ImageJ2 |
| ImageJ Biovoxxel plugin | http://www.biovoxxel.de/ | RRID:SCR_015825 |
| MoltiQ | https://imagej.net/plugins/motiq | https://doi.org/10.1091/mbc.e21-11-0585 |
| Zeiss Zen Black image acquisition software | https://www.micro-shop.zeiss.com/en/us/softwarefinder/software-categories/zen-black/ | RRID:SCR_018163 |
| GraphPad Prism | https://www.graphpad.com | RRID:SCR_002798 |
| **Other** | | |
| 2-micron Polybead Carboxylate Microspheres | Polysciences, Inc | Catalog #18327-10 |
| pHrodo™ iFL Red Microscale Protein Labeling Kit | Thermo Fisher | Catalog #P36014 |
| Alexa Fluor™ 555 Protein Labeling Kit | Thermo FIsher | Catalog #A20174 |
| 8 Well chambered cover Glass with #1.5 high-performance cover glass - 57 mm x 25 mm base | Cellvis | Catalog #C8-1.5H-N |

| Reagent/resource | Reference or source | Identifier or catalog number |
|---|---|---|
| 4 Well chambered cover Glass with #1.5 high-performance cover glass —57 mm × 25 mm base | Cellvis | Catalog # C4-1.5H-N |
| 0.75 mm biopsy punch | LabTech | Catalog #52-004908 |
| 35 mm Glass-bottom dish with 20 mm micro-well #1.5 cover glass | Cellvis | Catalog #D35-20-1.5 N |
| 3.5 mm biopsy punch | Robbins Instruments | Catalog #RBP-35 |
| Borosilicate glass capillaries | Kimble | Catalog #34500-99 |
| Horizontal Puller | Sutter Instrument | |
| Zeiss 7MP Microscope with pulse infrared laser (Chameleon Vision2) | Microscope – Zeiss Laser – Coherent | |
| Leica VT1200S Vibratome | Leica | |
| Olympus IX83 Microscope with an environmental chamber on a Tokai Hit INU incubation system controller | Olympus | |
| BioTek Cytation 5 Cell Imaging Multimode Reader Microscope | Agilent | |
| 40-μm sterile cell strainer (blue) | Fisher | Catalog # 22-363-547 |
| Millipore® Steritop® Vacuum Bottle Top Filter with 150 mL Funnel 45 mm threading | Millipore Sigma | Catalog # SC2GT01RE |

## Cells

Established murine microglial cell line BV-2s were used for all experiments listed. Cells were grown in complete cell culture media, containing Dulbecco's Modified Eagle Medium (DMEM) (Gibco, 31053028), 5% fetal bovine serum (FBS) (Gibco, 10437028), 1% glutaMAX (Gibco, 35050061), and 1x antibacterial-antimycotic (Gibco, 15240062) at 37 °C and 5% $CO_2$. About 70–80% confluent dishes were treated with 0.05% Trypsin-EDTA solution (Caisson Labs TRL02-100ML) at 37 °C for 10 min. The solution was then aspirated and replaced with 2 mL of complete cell culture media that was gently sprayed over the bottom of the dish to dislodge cells and collect for counting and passage into new dishes. BV-2 cells were tested for mycoplasma (Invivogen rep-mysnc-50) upon thawing of frozen stocks.

## Labeling iC3b with Alexa Fluor

Alexa Fluor 555 Fluorescent Protein Labeling Kit (Thermo Fisher, A20174) was used as directed to label 250 μM of Human iC3b (CompTech, A115). Western Blot analysis was performed with the Alexa Fluor 555 labeled iC3b (AF-iC3b), utilizing a Ponceau Stain to verify that the iC3b is labeled and there was no excess dye present in the solution.

## Adhesion assay

Eight-well chamber dishes (Cellvis, C8-1.5H-N) were prepared with a coating of 10 µg/mL fibronectin (Gibco, 33016015) or 20 µg/mL AF-iC3b, as described above, then all wells were filled with PBS. Confluent BV-2 dishes were handled, and cells were counted via a Hemocytometer. About 0.5 mL Eppendorf tubes were each filled with 100 µL of complete cell culture media and 40,000 cells. Tubes were spun down at 1000×g for 5 min, then media aspirated. Cells were resuspended in 47.5 µL of 2% FBS (diluted in PBS) and 2.5 µL of the desired 1 mg/mL antibody to create a 50 µg/mL antibody concentration. Antibodies used were Normal Rat IgG (Invitrogen, 31933), Integrin αM (or Cd11b) (Abcam, ab8878), and Integrin β2 (NovusBio, NBP1-41272). Tubes were placed at 37 °C for 10 min. Cells were spun down again and resuspended in 40 µL of complete cell culture media. The eight-well chamber dish was then prepped with 1 mL of media in each well, and half of the cells were transferred from each antibody tube to the corresponding well. Dishes were placed on a BioTek Cytation microscope, incubated at 37 °C, and imaged every 30 min for 2 h. Plates were then removed and received one wash of 1x Cell Culture Phosphate Buffered Saline (Corning, MT21040CV) (referred to from here on as PBS) before being placed back on the microscope and a final image was taken.

## Analysis of adhesion assay

Image files were opened in ImageJ. Using the Counting Cells plugin, the total cells per field of view, both pre- and post-wash were counted. Counts were transferred into GraphPad to graph. Graphs represent the percentage of pre-cells remaining, the percentage of lost cells in the wash, and the total number of cells present during the pre-wash stage.

## Four-chamber dish creation (iC3b coating phagocytosis)

Four-chamber well dishes were coated with fibronectin (10 µg/mL) or AF-iC3b (1 µg/mL, 10 µg/mL, or 20 µg/mL) for 1 h at 37 °C. Chambered dishes were washed three times with PBS. Wells were filled with PBS, and dishes were stored wrapped in parafilm for up to one month at 4 °C.

## Motility microscope assay

Upon completing the 3-h incubation, chambered dishes were placed in an environmental chamber on a Tokai Hit INU incubation system controller on the Olympus IX83 to be held at 37 °C, 90% humidity, and 5% $CO_2$ for the duration of the live cell imaging. A 20X air objective was employed during overnight time-lapse imaging. Ten fields of view per well were selected. Images were taken at each field of view in the relief contrast channel every 10 min for the span of 16 h.

## Analyzing motility

Each motility file was opened in FIJI ImageJ. The Manual Tracking plugin was used to track ten cells per field of view and combine data across all fields of view within treatment groups. Cell tracking was stopped if a cell divided or ran into another cell, resulting in a changed

direction. The resulting measurements were uploaded into the Ibidi Chemotaxis and Migration Tool ImageJ plugin to measure velocity, accumulated distance, and persistence. Persistence (also termed d/T) is a measure of the straight-line distance between a cell's starting and ending position on a track (d) divided by its overall track length (T). Thus, a value very close to zero (high T) is understood to be meandering much more than a cell moving along the shortest distance between its start and end (d = T), which has a value of 1. The closer the value is to 1, the more persistent the migration is. These data tables were exported to GraphPad Prism to graph differences in velocity, distance, and persistence between treatment groups.

## Analysis of iC3b uptake

Experimental images for the three AF-iC3b wells were loaded into ImageJ, and the relief contrast channel was removed. The BioVoxxel Pseudo Flat Field Correction plugin was used to remove uneven fluorescence from the Olympus IX83 capturing ability. Manual thresholding was performed to eliminate non-cell clearance detection. The image was made binary, and then the raw intensity difference was measured for the first and last time point in the stack. The change in raw intensity between the two timepoints quantified how much of the iC3b fluorescence was no longer visible, signifying uptake by the cells.

## Antibody treatment for iC3b coating uptake

4 well chamber dishes were coated with 10 µg/mL AF-iC3b. BV-2 cells then received antibody treatment as described above in the adhesion assay method. Cells were seeded into the dish and given 2 h to settle and spread before the dish was placed on the microscope and imaged for 16 h. Analysis compared masks of AF-iC3b clearance off the bottom of the dish at time 0 and after 16 h.

## Four-chamber dish creation (bead phagocytosis)

Four-well chamber dishes with two media wells and two agarose wells were prepared as previously described (Paulson et al, 2025). In short, four-well chamber dishes with #1.5 cover glass bottoms (Cellvis, C4-1.5H-N) were coated in 10 µg/mL fibronectin (Gibco, 33016015) for 1 h at 37 °C. Chambered dishes were washed three times with PBS. Two wells were each filled with 1 mL PBS and two wells were each filled with 1 mL 1% agarose. Agarose was prepared as follows: 2.5 mL of 2x Hank's Balanced Salt Solution (Sigma, H1387-10X1L) and 5 mL of complete cell culture media (as defined above) were combined and placed in a 68 °C bath for 1 h. About 0.12 g of low gelling temperature agarose (Sigma, A9045-5G) was added to 2.5 mL of sterile water (Corning, 25-055-CV) and heated in a microwave until fully dissolved. The two solutions were combined to create a warm 1% agarose mixture. One mL of the mixture was poured into each well and allowed to solidify for 90 min at room temperature. Chambered dishes were wrapped in parafilm and stored at 4 °C for up to 1 month.

## Phagocytosis bead creation

Label 100 µg Human iC3b (CompTech, A115) using pHrodo™ iFL Red Microscale Protein Labeling Kit (Thermo Fisher, P36014) according to manufacturer instructions. Precipitated labeled protein was opsonized

to 30 µL of 2-micron Polybead Carboxylate Microspheres (Polysciences, Inc., 18327-10) (hereafter referred to as beads) in 1 mL of 1x PBS at 4 °C overnight. Beads received three washes with 1x PBS after completing opsonization and were resuspended in a final volume of 500 µL, stored at 4 °C for up to 4 months. Before usage in an experiment, beads were vortexed vigorously.

## Cell preparation

Agarose chambered dishes were prepped for cells by replacing PBS with 1 mL complete cell culture media. Agarose wells had two punches placed in each for cell and bead insertion using a 0.75 mm biopsy punch tool (LabTech, 52-004908). Confluent dishes were treated according to the standard passaging protocol (above). For cell migration experiments, 13,000 cells were added to each media well. For phagocytosis, 18,000 cells were added to each media well. Lower numbers were used for migration experiments to minimize cell-cell interactions and allow more room for random migration uninterrupted. For both types of experiments in the confined condition, 15,000 cells were inserted under the agarose in each punched location using a gel-loading pipette tip. If the volume of media containing 15,000 cells exceeded 15 µL, the cells were spun down at 1000×g and resuspended in 10 µL of complete cell culture media. Chambered dishes were placed at 37 °C for 3 h to allow cells to settle and spread before being moved to the microscope. For the inhibitor trials, each cell culture media well received either vehicle or inhibitor to create the working concentration used for cell treatment (more information below).

## Phagocytosis microscope assay

Upon completing the 3-h incubation, iC3b beads were added to the wells. Media wells received 5 µL of beads. For confined wells, a 1:4 dilution of the beads was made with PBS and inserted into each punch in the agarose to match the bead density with the media condition. Chambered dishes were then placed in a Tokai Hit INU incubation system controller on the Olympus IX83 to be held at 37 °C, 90% humidity, and 5% CO$_2$ for the duration of the live cell imaging. A 20X air objective was employed during overnight time-lapse imaging. Fifteen fields of view per well were selected. Images were taken at each field of view every 10 min for the span of 8 h. Both the relief contrast and the DsRed channels were utilized. The DsRed channel was set to 500 ms exposure to detect pHrodo signal upon bead internalization by cells.

## Analyzing bead density

Each phagocytosis file was opened in Fiji ImageJ. A bandpass filter was then run on the relief contrast channel. Structures were filtered to fit between 5 and 15 pixels and then threshold filtered until only the beads were highlighted in red. The analyze particles function was used to count bead density. Area threshold was set to 10–30 micron$^2$, circularity threshold was set to 0.8–1.0, and outlines were shown. When summarized, average size was maintained between 15 and 17 micron$^2$ across the time lapse, with the threshold being adjusted and analysis rerun if the averages were too small or too large. Counts of beads per time point were averaged to create the bead density designator for each file. Averaged bead numbers falling between 50 and 500 were designated low bead density; between 500 and 950 medium bead density; between 950 and 1550 high bead density.

## Analyzing phagocytic rate

Each phagocytosis file was opened in FIJI ImageJ. Channel colors were corrected, and brightness/contrast adjusted to create the same brightness and contrast parameters for each file. Using the cell counter plugin, the total number of fluorescent and non-fluorescent cells was counted at each measured time point throughout the phagocytosis videos. Comparisons between treatment groups at individual timepoints, as well as comparing changes over time within each group, were conducted with GraphPad Prism. Graphs were normalized to T = 0. Fluorescent cells at the beginning of the videos were removed from the count of both fluorescent cells and total cells per field of view. Data comparisons were broken up by bead density, only comparing media and confined fields in the low bead density category to each other.

## Analyzing phagosome size

Each phagocytosis file was opened in CellSens Dimension software under the Count and Measure plugin. Under detection options, the minimum object size was set to 5 pixels. Under the Thresholding tab, the adaptive threshold was selected. Selecting the DsRed channel, the maximum was set to 65,000 (the maximum) and the minimum was changed for each image to label only internalized beads, although never going below 1000. Selecting the relief contrast channel, the maximum was set to 2 and the minimum to 1 so that only fluorescent labels were detected. Outputs of the total number of phagosomes and their average size per field of view were then verified at each timepoint before the data were exported to an excel file. These data were presented in three ways: (1) The average size of phagosomes was reported by dividing the average size per field of view at 2 h by the total number of phagosomes detected at 2 h. (2) The average size of phagosome per phagocytic cells was reported by dividing the average size per field of view by the number of phagocytic cells at that timepoint. (3) The average number of phagosomes per phagocytic cell was reported by dividing the total number of phagosomes at 2 h by the number of phagocytic cells at that timepoint. The average size of the phagosome and the average size of phagosome per phagocytic cell differ by examining if beads are trafficked into the same or different lysosomes, and the latter examines if more beads overall are being trafficked in each treatment condition.

## Antibody treatment for iC3b bead uptake

Four-well chamber dishes were coated with 10 µg/mL FN. BV-2 cells then received antibody treatment as described above in the adhesion assay method. Cells were seeded into the dish and given 2 h to settle and spread, before moving to the microscope and imaging for 2 h. Analysis for percent fluorescent cells was performed as described above.

## Four-chamber dish creation (iC3b coating with CK-666 phagocytosis)

The small molecule Arp2/3 complex inhibitor CK-666 (Abcam, ab141231) was resuspended in anhydrous DMSO (Thermo Fisher, D12345) to create a stock solution. Four-chamber well dishes were coated with 10 µg/mL of AF-iC3b for 1 h at 37 °C. Chambered

dishes were washed three times with PBS. Agarose was prepared as detailed above and then split into two aliquots of 5 mL warm agarose solution. Each aliquot received either vehicle (anhydrous DMSO) or CK-666 at a final concentration of 125 μM. AF-iC3b-coated dishes now contained two PBS wells, one vehicle agarose well, and one CK-666 agarose well. Chambered dishes were wrapped in parafilm and stored at 4 °C for up to 1 month. Cells were seeded, incubated, imaged, and analyzed similarly to non-CK-666-treated runs.

## Four-chamber dish creation (bead phagocytosis with CK-666)

About 125 μM CK-666 or an equivalent volume of DMSO was added to cooled 1% agarose and mixed by swirling just prior to addition to chambers, as previously described (Paulson et al, 2025). Before agarose addition, each chamber was coated with 10 μg/mL FN. Chambered dishes were wrapped in parafilm and stored at 4 °C for up to one month. Cells were seeded, incubated, given pHrodo-iC3b beads, imaged, and analyzed similarly to non-CK-666-treated runs.

## Haptotaxis agarose plate preparation

Haptotaxis dishes were created as previously described (Stinson et al, 2025). In short, 10 mL of a cell media solution composed of 70% complete cell culture media and 30% L929-conditioned media (for a source of M-CSF) was combined with 5 mL of 2x Hank's Balanced Salt Solution (Sigma, H1387-10X1L) and placed in a 68 °C bath for 1 h. 0.24 g of low gelling temperature agarose (Sigma, A9045-5G) was added to 5 mL of sterile water (Corning, 25-055-CV) and heated in a microwave until fully dissolved. The two solutions were combined to create a warm 1% agarose mixture. About 3 mL of this mixture was poured into a 35 mm glass-bottom dish containing a 20 mm micro-well (Cellvis, D35-20-1.5-N) and allowed to solidify for 90 min at room temperature. The plates were not pre-coated with substrate. Plates were wrapped in parafilm and stored at 4 °C for up to 1 month, after which point the gel's integrity could be compromised.

## Haptotaxis cell and dish preparation

A 3.5 mm biopsy punch tool (Robbins Instruments, RBP-35) was used to generate wells in the agarose gel. Punches were created, and then excess gel was aspirated to create each well. A center well was created, and AF-iC3b was allowed to spread under agarose from the central well for 1 h at 37 °C. The well was washed once with PBS, and then two linear wells of 1.25 mm well spacing were punched in the agarose. Cells were prepared and counted via a hemocytometer. Once counted, 100,000 cells were transferred to two separate Eppendorf tubes (200,000 cells total) and spun down at $1000 \times g$ for 3.5 min. Excess media was aspirated, and cells were resuspended in 20 μL of cell culture media. These suspensions were then transferred to each of the cell wells in the prepared agarose dish.

## Haptotaxis migration imaging

Cell plates were placed on the Olympus IX83, and an INU incubation system controller from Tokai Hit was used to maintain cells in a stable environment at 37 °C, 90% humidity, and 5% $CO_2$. Under-agarose chambers were placed into the environmental chamber. A 20X air objective was employed during overnight time-lapse imaging. Cells were then imaged via relief contrast and the DsRed channel. A total of 32 positions were selected to assay macrophage haptotaxis for each experiment: four non-overlapping sites from the top to the bottom of the edge of a cell well, which composed Zone 1, and four more non-overlapping sites at the same Y position but shifted ~665 μm (1 field of view) away, which composed Zone 2. This process was repeated on the other side of the same well, and then on both sides of the second cell well. An additional 12 images were taken across the length of the chamber between the (+) regions of the cell wells for fluorescence intensity analysis. Images were captured at 10-min intervals for 24 h.

## Haptotaxis migration analysis

Each of the Zone 2 files were opened in ImageJ. The Manual Tracking plugin was used to track each cell per field of view and combining data across fields of view within that file's Zone 2 set of 4 files. Cell tracking was stopped if a cell divided or interacted with another cell, resulting in a changed direction. The resulting measurements were uploaded into the Ibidi Chemotaxis and Migration Tool ImageJ plugin to measure track velocity, accumulated distance, persistence, and forward migration index in the X coordinate (FMIx). FMIx is calculated by dividing the difference between the starting and ending X coordinate of a cell track value by the accumulated distance. These data tables were exported to GraphPad Prism to graph differences in velocity, distance, and persistence between treatment groups. Note that persistence is not the same as FMIx, as the persistence values do not consider any external frame of reference (e.g., an external gradient).

## Alexa Fluor 555 iC3b (AF-iC3b) quantification

AF-iC3b was allowed to spread under agarose from the central well for 1 h at 37 °C prior to each migration assay. Twelve images were taken across the length of the chamber between the (+) regions of the cell wells during migration runs. Each position was opened in ImageJ, and the DsRed channel was isolated. A $500 \times 1$ pixel line is created through the center of the image to measure the fluorescence intensity. A Gray value plot was generated, and the data were saved to Excel and then loaded into Graphpad Prism. This process was similarly done for the soluble iC3b quantification, which involved plates receiving 0, 1, 2, or 5 washes of PBS before imaging and then quantification.

## CK-666 or CK-689 under-agarose haptotaxis assay

Haptotaxis agarose dishes were prepared, with the addition of either 125 μM CK-666 or 125 μM CK-689 to the final gel volume as previously described (Stinson et al, 2025). About 3 mL of this mixture was added to each 35 mm dish. The agarose chamber was then prepared as described above with one source well and two cell wells. About 20 μL of 250 μg/mL AF-iC3b was added to the source well and allowed to diffuse for 1 h at 37 °C. After 1 h, the source well was washed once with PBS. Cells were dissociated, counted via hemocytometer, and seeded into the plate. Imaging and analysis were conducted as described above.

## Hippocampal slice preparation

Parental CX3CR1$^{GFP/GFP}$ mice were purchased from Jackson Labs (B6.129P2(Cg)-*Cx3cr1*$^{tm1Litt}$/J; JAX strain 005582) (Jung et al, 2000). These mice were bred to C57Bl6/J (JAX strain 000664) mice and the resulting CX3CR1$^{GFP/+}$ mice were used in experimentation. Mice were housed in a group setting and received food and water ad libitum. Conditions were monitored by the lead author, and trained facility staff according to institutional care and use directives. Active USUHS IACUC-approved mouse protocols covered all animal experimentation for this study, in compliance with ethical guidelines. Equal numbers of male and female mice were used in these experiments. Between 35 and 50 days of age, mice were sacrificed by $CO_2$ inhalation. All mice were immuno-competent and of normal weight. They were not tested, drug-treated or subject to any procedures prior to tissue harvesting. 400-µm thick coronal sections were cut in 4 °C dissection solution (high-sucrose artificial cerebrospinal fluid (aCSF) containing (in mM): KCl, 2; $NaH_2PO_4 \cdot H_2O$, 1.25; $MgSO_4 \cdot 7H_2O$, 2; $MgCl \cdot 6H_2O$, 1; $NaHCO_3$, 26; $CaCl_2 \cdot 2H_2O$, 1; D-glucose, 10; sucrose, 206; bubbled with a mixture of 95% $O_2$/5% $CO_2$) on a Leica VT1200S Vibratome (Buffalo Grove, IL, USA) and subsequently incubated for 30 min at 37 °C in normal aCSF (containing (in mM): NaCl, 126; KCl, 3, $NaH_2PO_4 \cdot H_2O$, 1.25; $MgSO_4 \cdot 7H_2O$, 2; $NaHCO_3$, 26; $CaCl_2 \cdot 2H_2O$, 2; D-glucose, 10; sucrose, 20; bubbled with a mixture of 95% O2/5% $CO_2$).

Brain slices were kept at room temperature for up to 8 h. For imaging, the slices were mounted into a tissue slice chamber (Warner Instruments) and perfused with normal aCSF at room temperature. A ZEISS 7MP microscope was used to perform two-photon fluorescence microscopy ~80 µm below the surface of the slice. The microscope system used a pulsed infrared laser (Chameleon Vision2, Coherent, Santa Clara, CA) with a tunable wavelength range from 680 to 1080 nm and was set to 850 nm for the time series experiments and 800 nm for the overview images, respectively. Fluorescence was detected with three non-descanned detectors (NDDs) equipped with a 480 nm short pass for blue detection (dextran), a 525/50 nm bandpass for green detection (GFP) and a 605/70 nm bandpass for red detection (pHrodo-iC3b-labeled beads). Imaging was done using a 40x water immersion objective (NA = 1.0) and controlled with Zeiss ZEN image acquisition software.

All imaging was conducted in the CA1 region of the hippocampus. For surveillance experiments, 20–40 frame stacks (1 µm step size, dependent on cell size) were acquired at a frequency of one stack per 1 min for a total duration of 10 min. For chemotaxis, 61 frame stacks (1 µm step size) were acquired at a frequency of one stack per 2 min for a total duration of 60 min. For iC3b-bead interaction experiments, 61 frame stacks (1 µm step size) were acquired at a frequency of one stack per 2 min for a total duration of 120 min.

For stimulus application, glass pipettes were fabricated from borosilicate glass capillaries on a horizontal puller (Sutter Instrument, P-80/PC) and fire-polished to a final tip diameter of ~2 µm for chemotaxis experiments and 5 µm for bead application. Injections contained Cascade Blue labeled 3000 MW Dextran (Thermo Fisher, D7132), aCSF, and 1 mM ATP for chemotaxis experiments or dextran, aCSF, and pHrodo-iC3b opsonized 2 µm beads for iC3b sensing experiments. For drug treatment, brain slices were incubated in aCSF containing 200 µM of CK-666 or the equivalent amount of DMSO for 2 h prior to being placed into the tissue slice chamber. Drug was not added to the aCSF flowing through the system during imaging, but was added to the solution in the glass pipette for injection at the same concentration as incubation. Surveillance videos did not contain a stimulus application.

## Surveillance morphology analysis

Microglia skeletons and morphometrics were prepared using the automated plugin MotiQ (Hansen et al, 2022). Original images were first binarized using the mean algorithm, then the close function was used to capture incomplete processes as continuous. MotiQ cropper was applied when appropriate, and thresholder (with no preprocessing or stack processing) and 2D analyzer (no Gauss filter, remove all particles but the largest) were used to generate skeletons and morphology metrics. Poorly masked cells or timepoints were excluded from analysis.

## Surveillance area analysis

Total cell area: Cell area was determined using manual thresholding (default settings) due to variable brightness between images. The threshold was decided based on the inclusion of processes while minimizing background, and the measure function was used to capture the thresholded area. Cell body area: Area was calculated using a line trace drawn around the cell body and measured using the measure function in Fiji ImageJ. Proportion of total cell area was calculated by dividing the cell body area by the total cell area.

## Chemotaxis analysis

Images were measured using Fiji ImageJ to detect the length of process travel in maximum projection videos between the injection time point and reaching the pipette tip. This same process measured how far processes traveled up the inside of the pipette. Process motility was tracked similarly to cell motility tracking defined earlier.

## Bead interaction analysis

Beads were counted by hand for color overlap at each end time point. Processes interacting with the beads were measured using ImageJ. Process motility was tracked similarly to cell motility tracking defined earlier.

## Study design

Cells or tissue slices were assessed visually for health (normal cell morphology and/or tissue integrity) and then randomly assigned to control or experimental groups. The sample size was not explicitly set using power analysis. Researchers were not blinded to the sample groups.

## Statistical analysis

The Kruskal–Wallis with Dunn multiple comparisons test was used to assess significance in experiments where a normal distribution of the dataset could not be assumed. When only two experimental

conditions were tested, we used Mann–Whitney tests when we could not assume normality. Unpaired *t*-tests and ANOVAs were used when normality tests indicated a normal distribution of the data. In addition, we have prepared an appendix containing effect sizes for each major cell-based finding in the current study. Effect sizes were calculated using the mean of all experimental means ("mean of means"), and both Cohen's d and Hedges' g values are reported. Hedges' g is likely the more relevant value given the experimental sample size. All statistics, besides effect sizes, were calculated using GraphPad Prism, and significance was assumed if $p \leq 0.05$. Size effects were calculated via Excel and are included both as source data in the Excel format and as a dataset. More information on each statistical test can be found in the relevant figure legend panel.

## Graphics

Graphics created and used in the synopsis, panels A and G of Fig. 3, and panel A of Fig. EV2 were created using BioRender.com.

## Data availability

No large primary datasets have been generated and deposited.

The source data of this paper are collected in the following database record: biostudies:S-SCDT-10_1038-S44319-026-00720-9.

## Peer review information

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

## Acknowledgements

We thank the members of the Rotty Lab for helpful discussions during research development. We thank Dr. Zygmunt Galdzicki for the use of his lab's vibratome. We thank Dr. Barrington Burnett for the BV-2 cells. We thank the Biomedical Instrumentation Center for access to the Zeiss LSM 7MP two-photon microscope. This project is sponsored by the Uniformed Services University of the Health Sciences (USU); however, the information or content and conclusions do not necessarily represent the official position or policy of, nor should any official endorsement be inferred on the part of, USU, the Department of Defense, or the US Government. This work was supported by a Uniformed Services University graduate student research award (to SGP), a Cosmos Club Foundation award (to SGP), and by the National Institutes of Health (GM134104, to JDR), Department of Defense (HU00012320103, to JDR), and startup funds from the Uniformed Services University (to JDR). The Uniformed Services University of the Health Sciences (USU), 4301 Jones Bridge Rd., A1040C, Bethesda, MD 20814-4799, is the awarding and administering office.

## Author contributions

**Summer G Paulson**: Conceptualization; Data curation; Formal analysis; Validation; Investigation; Visualization; Methodology; Writing—original draft; Writing—review and editing. **Isabella Swafford**: Formal analysis; Methodology; Writing—original draft; Writing—review and editing. **Fritz W Lischka**: Formal analysis; Methodology; Writing—review and editing. **Jeremy D Rotty**: Conceptualization; Supervision; Funding acquisition; Methodology; Writing—original draft; Project administration; Writing—review and editing.

Source data underlying figure panels in this paper may have individual authorship assigned. Where available, figure panel/source data authorship is listed in the following database record: biostudies:S-SCDT-10_1038-S44319-026-00720-9.

## Disclosure and competing interests statement

The authors declare no competing interests.

# Expanded View Figures

**Figure EV1. Confirmation of Alexa Fluor 555 labeling of iC3b.**

(**A**) Ponceau Stain of 8 μg of AF-iC3b, either before spinning down or after a 60 s spin down. (**B**) Example images of cells plated on FN or AF-iC3b treated with FBS (vehicle control), IgG (negative control), αM, or β2 pre- and post-wash for adhesion assay. Scale bar represents 250 μm. (**C**) Percent of cells lost in the process of washing wells in the adhesion assay. (**D**) Percent of cells from pre-washing remaining in post-wash images. (**E**) The total number of alive cells in the pre-wash images. (**F**) Normalized mean difference in intensity of iC3b clearance from the bottom of 10 μg/mL iC3b-coated dishes by BV-2 cells treated with FBS, IgG, αM, or β2 antibodies. Normalization was done to the FBS condition. (**G**) Percent phagocytosis of iC3b-opsonized beads in BV-2 cells at 2 h, after treatment with CR3 blocking antibodies or normal IgG. $N = 3$ experiments (biological replicates) for the antibody adhesion assay (**C–E**), $N = 2$ experiments (biological replicates) for the iC3b uptake with antibody treatment (**F**), and $N = 3$ experiments (biological replicates) for the phagocytosis with antibody assay (**G**). For (**C–E**), statistical analysis was assessed using two-way ANOVA tests. For (**G**), statistical analysis was assessed using ordinary one-way ANOVA. Error bars represent the mean and SEM in all graphs. Values for $p$ are as follows: (**C**) ns = nonsignificant, *$p = 0.0108$. (**D**) ns nonsignificant, **$p = 0.0097$. (**E**) ns nonsignificant. (**G**) ns nonsignificant, ****$p < 0.0001$, *$p = 0.0396$.

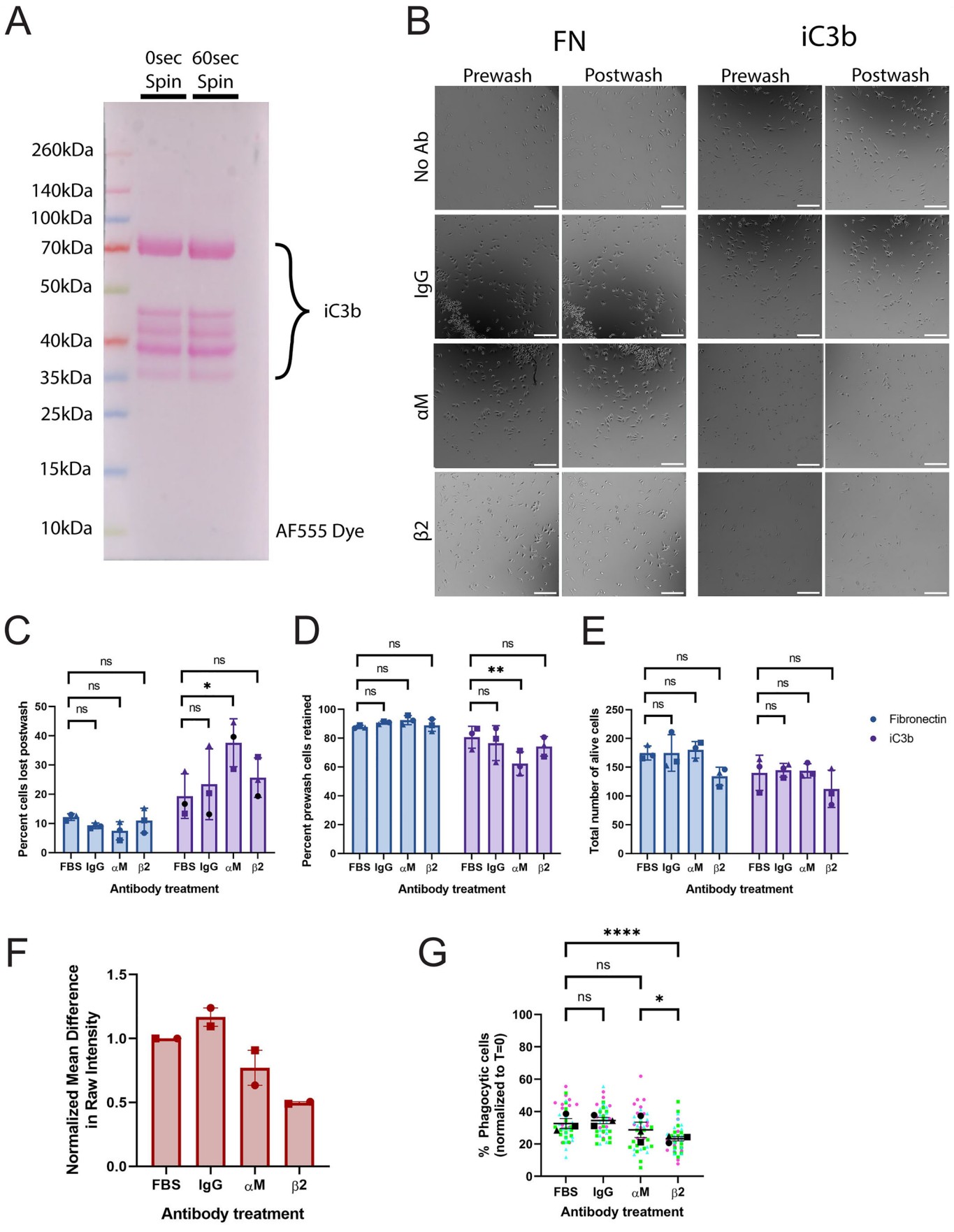

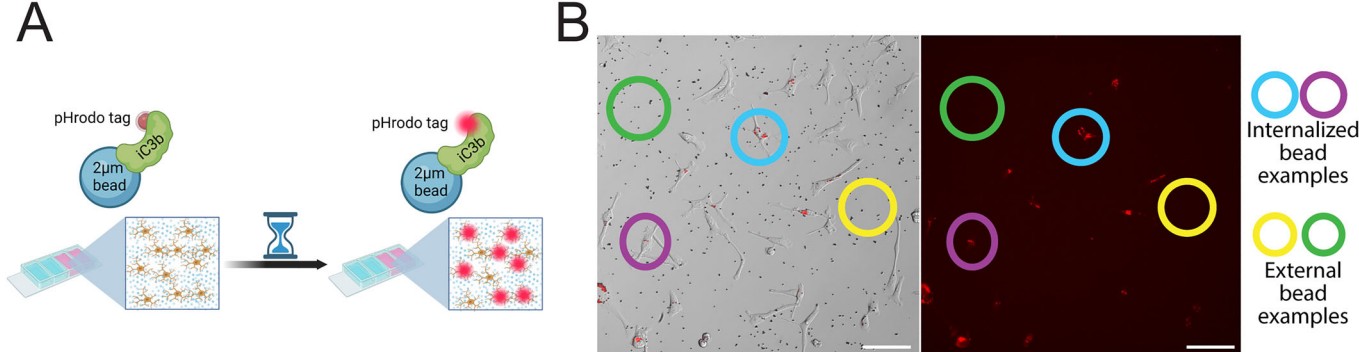

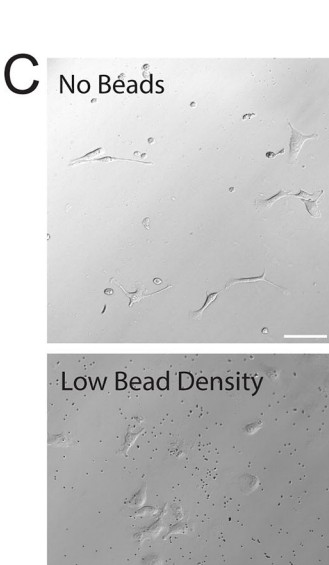

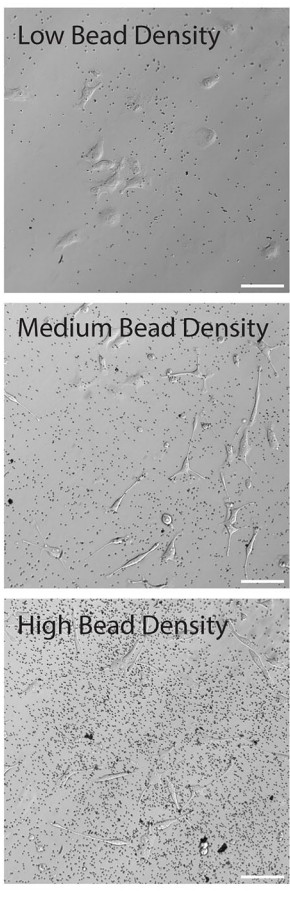

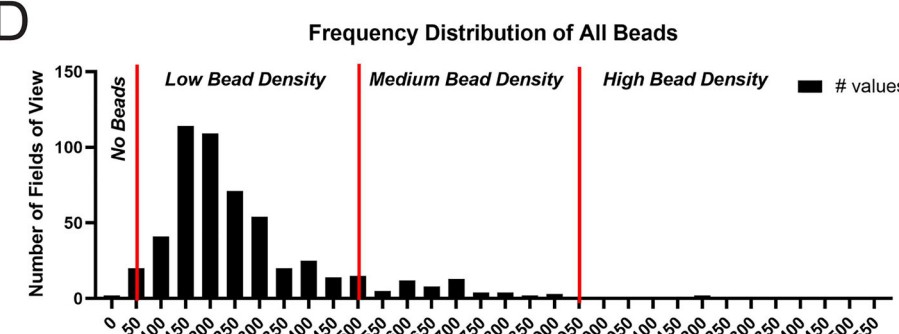

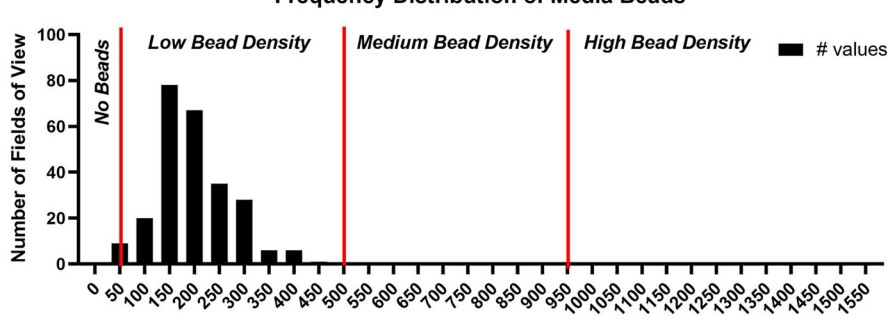

**Figure EV2. Strategy to control for bead density.**

(A) Schematic depicting phagocytic bead labeling and the pHrodo tag fluorescing inside cells but not externally. (B) Duplication of Fig. 1J. Examples of either internalized (blue and purple circles) or external (yellow and green circles) pHrodo-red staining outlined in both composite and pHrodo only images. Scale bar represents 100 μm. (C) Example phase contrast images displaying different bead densities. No beads (top) through high bead density (bottom). Scale bar represents 100 μm. (D) Histograms detailing the breakdown of average bead densities per field of view across all experimental runs. Low bead density was classified as any field of view with a bead average between 50 and 500 beads; medium bead density was 500 to 950 beads; high bead density was any field of view average above 950 beads. Low bead densities in confined images most closely matched the density of typical media images.

 

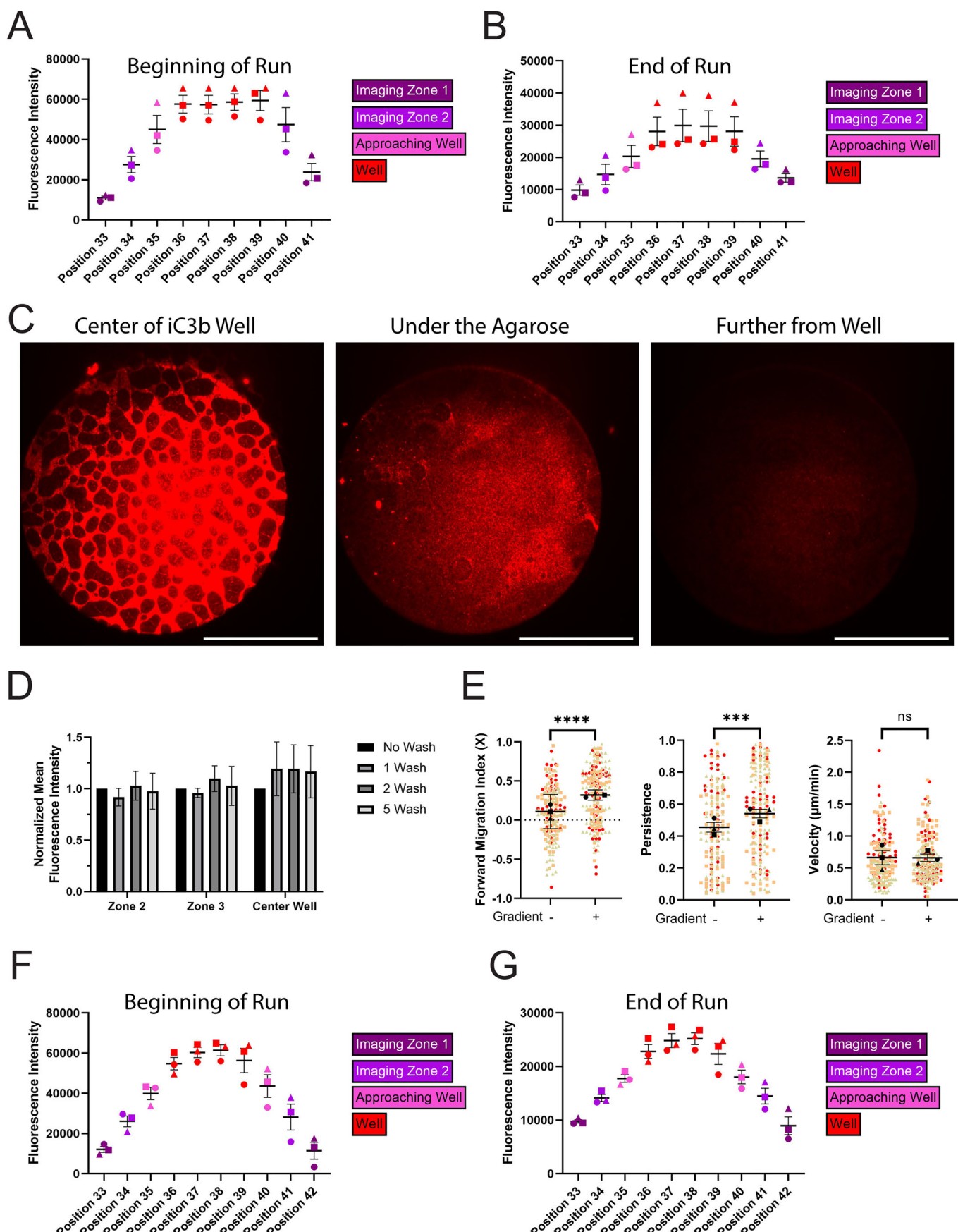

**Figure EV3.  Examining consistency of AF-iC3b labeling during haptotactic assays.**

(A, B) Measurement of the mean fluorescent intensity during haptotaxis runs of the AF-iC3b label spanning from the Zone 1 of one cell well across the center well to the Zone 1 section of the second well. Measurements are color-coded by position (see legend), and symbols represent the three haptotaxis runs. (A) corresponds to the beginning of the runs, and (B) corresponds to the 24-hour point of the run. (C) TIRF images of the glass bottom of the haptotaxis apparatus to demonstrate the binding of AF-iC3b to the glass and the gradient as the signal diminishes, the further from the center well the image is taken. Scale bars represent 50 μm. (D) Normalized iC3b mean fluorescence intensity for dishes that received 0 washes, 1 wash, 2 washes, or 5 washes with PBS after 1 h of iC3b coating. Positions imaged are relative to the imaged zones from the cell well. (E) Measurements for FMIx (left), persistence (center), and velocity (right) for cells migrating during 125 μM CK-689 treatment. (F, G) These graphs are the same as (A, B), but with the CK-666 haptotaxis runs. $N = 3$ experiments (biological replicates) for all graphs; $n = 60$ cells per experiment for (E). Values for (A, B, D, F, G) were taken at 1 image per position. Statistical analysis was assessed using the Mann–Whitney test. Error bars represent means and SEM in all graphs, except for (E) left, which is the mean and 95% CI (E center and E right are SEM). Values for $p$ are as follows: E left) ****$p < 0.0001$. E center) ***$p = 0.0004$. E right) ns nonsignificant.

    

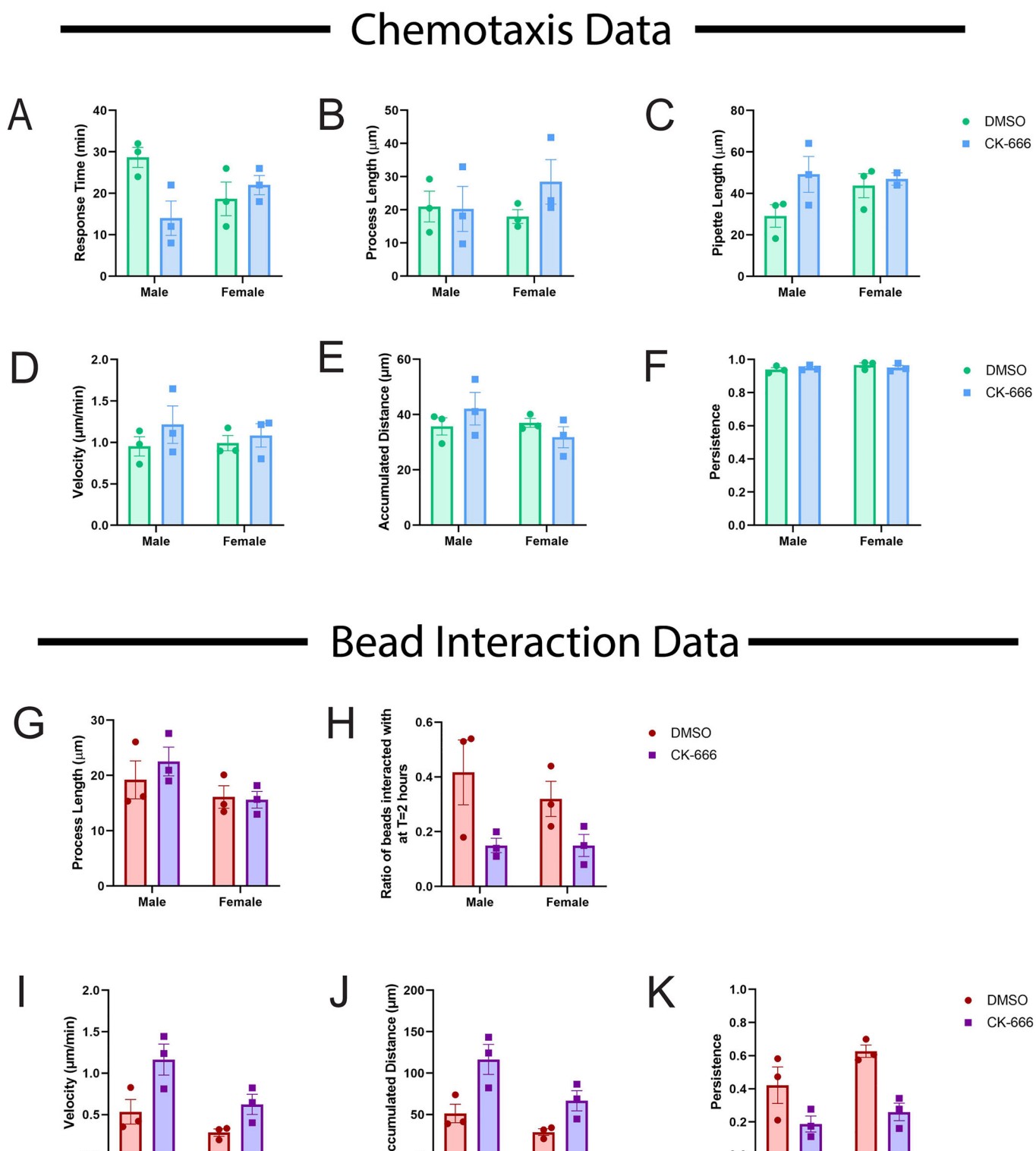

**Figure EV4. Lack of microglial sex differences in situ in response to Arp2/2 complex inhibition.**

(A–C) ATP results of (Fig. 5B-D) broken down into male and female specific results to examine sex differences. (D–F) ATP results of (Fig. 5E–G) broken down into male and female specific results to examine sex differences. (G, H) Results from (Fig. 6C, D) were broken down into male and female mice to examine sex differences. (I–K) Results of (Fig. 6E–G) are broken down into male and female specific results to examine sex differences. $N = 6$ experiments for each graph (3 male; 3 female biological replicates). Colors for vehicle or CK-666 treatment are clearly marked. Error bars represent SEM in all graphs.

