## [Peer Review File · EMBO Reports]

The Arp2/3 complex is required for in situ haptotactic response of microglia to iC3b

Summer Paulson, Isabella Swafford, Fritz Lischka, and Jeremy Rotty

Corresponding author(s): Jeremy Rotty (Jeremy.Rotty@usuhs.edu)

Review Timeline:

Submission Date:	18th Jun 25
Editorial Decision:	25th Jul 25
Revision Received:	23rd Oct 25
Editorial Decision:	19th Dec 25
Revision Received:	9th Jan 26
Accepted:	2nd Feb 26

Editor: Achim Breiling

Transaction Report:

Dear Dr. Rotty,

Thank you for the submission of your manuscript to EMBO reports. I have now received reports from the three referees that were asked to evaluate your study, which can be found at the end of this email. As you will see, the referees think that these findings are of high interest. However, they have several comments, concerns, and suggestions, indicating that a major revision of the manuscript is necessary to allow publication of the study in EMBO reports. As the reports are below, and all the referee concerns need to be addressed, I will not detail them here.

Given the constructive referee comments, I would thus like to invite you to revise your manuscript with the understanding that the concerns of the referees must be addressed in the revised manuscript and/or in a detailed point-by-point response, as indicated in your revision plan. Acceptance of your manuscript will depend on a positive outcome of a second round of review. It is EMBO reports policy to allow a single round of revision only and acceptance of the manuscript will therefore depend on the completeness of your responses included in the next, final version of the manuscript.

1) a .docx formatted version of the final manuscript text (including legends for main figures, EV figures and tables), but without the figures included. Figure legends should be compiled at the end of the manuscript text.

2) individual production quality figure files as .eps, .tif, .jpg (one file per figure), of main figures and EV figures. Please upload these as separate, individual files upon re-submission.

4) a complete author checklist, which you can download from our author guidelines

(<https://www.embopress.org/page/journal/14693178/authorguide>). Please insert page numbers in the checklist to indicate where the requested information can be found in the manuscript. The completed author checklist will also be part of the RPF.

5) that primary datasets produced in this study (e.g. RNA-seq, ChIP-seq, structural and array data) are deposited in an appropriate public database. If no primary datasets have been deposited, please also state this in a dedicated section (e.g. 'No

primary datasets have been generated and deposited'), see below.

The accession numbers and database should be listed in a formal "Data Availability" section that follows the model below. This is now mandatory (like the COI statement). Please note that the Data Availability Section is restricted to new primary data that are part of this study. This section is mandatory. As indicated above, if no primary datasets have been deposited, please state this in this section

Data availability

6) We now request the publication of original source data with the aim of making primary data more accessible and transparent to the reader. You will receive a separate email with instructions for providing source data with your revised manuscript, including information how to upload and organize the files.

8) Regarding data quantification and statistics, please make sure that the number "n" for how many independent experiments were performed, their nature (biological versus technical replicates), the bars and error bars (e.g. SEM, SD) and the test used to calculate p-values is indicated in the respective figure legends (also for EV and Appendix figures). Please also check that all the p-values are explained in the legend, and that these fit to those shown in the figure. Please provide statistical testing where applicable. Please avoid the phrase 'independent experiment', but clearly state if these were biological or technical replicates. Please also indicate (e.g. with n.s.) if testing was performed, but the differences are not significant. In case n=2, please show the data as separate datapoints without error bars and statistics. See also: <http://www.embopress.org/page/journal/14693178/authorguide#statisticalanalysis>

9) Please add scale bars of similar style and thickness to all microscopic images, using clearly visible black or white bars (depending on the background). Please place these in the lower right corner of the images themselves. Please do not write on or near the bars in the image but define the size in the respective figure legend.

10) Please also note our reference format:

12) We now use CRediT to specify the contributions of each author in the journal submission system. CRediT replaces the author contribution section. Please use the free text box to provide more detailed descriptions and do NOT provide your final manuscript text file with an author contributions section. See also our guide to authors: <https://www.embopress.org/page/journal/14693178/authorguide#authorshipguidelines>

13) All Materials and Methods need to be described in the main text using our 'Structured Methods' format, which is required for all research articles. According to this format, the Methods section should include a Reagents and Tools Table (listing key

reagents, experimental models, software, and relevant equipment and including their sources and relevant identifiers), uploaded as separate file, and a Methods section in which we encourage the authors to describe their methods using a step-by-step protocol format with bullet points, to facilitate the adoption of the methodologies across labs. More information on how to adhere to this format as well as downloadable templates (.doc) for the Reagents and Tools Table can be found in our author guidelines (section 'Structured Methods'):

14) Please order the manuscript sections like this, using only these names:

Title page - Abstract - Keywords - Introduction - Results - Discussion - Methods - Data availability section - Acknowledgements (please put here all the funding information) - Disclosure and Competing Interests Statement - References - Figure legends - Expanded View Figure legends

15) Please make sure that all the funding information is also entered into the online submission system and that it is complete and similar to the one in the acknowledgement section of the manuscript text file.

16) Please name the movie files using 'Movie EVx' ('Movie EV1', 'Movie EV2', 'Movie EV3' ...) in all places (source file names, titles in the submission system, their callouts and legends). Please provide each legend as a readme.txt file that then should be ZIPped up with its corresponding movie and uploaded (so that we have one zip folder per movie). Finally, please remove the movie legends from the manuscript main text file.

I look forward to seeing a revised form of your manuscript when it is ready.

Yours sincerely,

Referee #1:

The paper is well-written, and the message is clear: microglial responses to iC3b occur in an Arp2/3 complex-dependent manner. Both iC3b-induced haptotaxis and phagocytosis of iC3b-coated beads are disrupted by Arp2/3 inhibitor treatment. This is further supported by the observation that the directional movement of microglia toward an iC3b gradient is abolished in the presence of CK-666. Additionally, the authors examine the role of Arp2/3 in microglia in situ, demonstrating that Arp2/3 contributes to maintaining microglial morphology and responsiveness to iC3b. This information advances our understanding of the molecular mechanisms regarding microglial haptosensing.

In my opinion there are only minor points need to be improved:

1. The label on the scale bar is too small to read, e.g., in Fig. 1D, 1J, 2E, 2J, 5A, 6A. Please review and correct this issue throughout the paper.
2. In line 335, the authors should rephrase the phrase 'it worth nothing...'. They can simply describe the observed trend, which is consistent with the pooled data. Moreover, the trends for both males and females appear similar. Statistical tests on these samples are unnecessary and uninformative, even misleading, as the sample sizes are too small to yield reliable conclusions. The same applies to Fig. 6E, F, J, K, and L.
3. It has been reported that the sensitivity of the Arp2/3 complex to inhibitors can vary depending on its subunit composition. For example, ArpC1A-containing Arp2/3, which is more sensitive to CK-666, is predominantly found in neurons, whereas ArpC1B-containing Arp2/3, which can still generate actin branches in the presence of CK-666, is more common in immune cells. Do the authors have any information regarding the expression levels of Arp2/3 subunit isoforms in microglia? If so, it would be worthwhile to include this in the discussion.

4. Fig 5A. I am not sure I understand what I'm looking at, and it's unclear what I expect to see. Additional explanation would likely help clarify this.

Referee #2:

In this article, Paulson et al. investigate the role of the Arp2/3 complex in the interaction of microglia with iC3b-bound surfaces or phagocytic targets. In the first part of the paper, they use a microglia-like cell line to characterize cell migration and phagocytosis in vitro. They show that these cells migrate on iC3b-coated surface, although slowly and less persistently than on fibronectin. Pharmacological inhibition of Arp2/3 reduces migration in both unconfined and confined conditions and prevents the detachment of iC3b from the surface. Furthermore, they observed that Arp2/3 inhibition reduced the phagocytosis of iC3b-coated beads, both in confined and unconfined conditions. Next, they look at the effect of a gradient of iC3b on microglia migration under confinement. Microglia migrated both towards and away from the gradient. However, their persistence was lower and velocity higher when migrating away from the iC3b gradient. These differences were abolished by Arp2/3 inhibition. Finally, they use brain tissue sections to investigate the migration of primary microglia in a physiological environment. Unsurprisingly, they found a role of Arp2/3 in the ability of microglia to form protrusions and maintain a ramified morphology. Yet, Arp2/3 inhibition had no significant effect in microglia chemotaxis towards ATP. However, microglia could not form stable interactions and internalize iC3b-coated beads when Arp2/3 was inhibited.

These results support the assertion that Arp2/3 plays an important role in iC3b phagocytosis by microglia, as previously demonstrated in macrophages by the senior author. This is likely due to a critical role of Arp2/3 in establishing strong binding to iC3b, as suggested by both in vitro and in situ experiments. However, several assertions made in the manuscript are not sufficiently supported by experimental evidence. The author claimed that phagocytosis under confinement in an Arp2/3 dependent manner, yet, when Arp2/3 is inhibited, phagocytosis appears to be in confined versus unconfined conditions. The authors also state that haptotactic migration toward increasing iC3b concentration is Arp2/3 dependent. However, there is no statistical comparison of DMSO versus CK-666 treatment in Figure 3 and, as far as I can tell, Arp2/3 inhibition seems to only affect cells migrating away from the source of iC3b. Finally, the authors link the inability of microglia to bind to iC3b-coated beads in tissues to a haptotaxis defect. Since there is no gradient of iC3b on the beads, the experiment does not support this interpretation.

Additional comments:

In several figures, statistical analysis is not properly conducted. In general, statistics should compare various treatments or treated versus untreated cells within the same condition. For example, Figure S1C-E should compare the effect of different antibodies when for cells migrating on fibronectin on one hand, and cells migrating on iC3b on the other hand. This is critical for the interpretation of the data. In this instance, blocking beta2 does not seem to affect binding to iC3b. Likewise, in Figure 3, data from CK-666 treated cells should be compared to data from untreated cells.

Method descriptions lack clarity. In several places, the mention "as previously described" is not associated with any reference. In these cases, the authors should provide complete descriptions of their procedures. The description of the phagocytosis assay under confinement lacks clarity. Are beads added to the same punch? If so, how far from the punch images were acquired for quantification and what criteria were utilized to determine that distance?

The authors use carboxylated beads coated with iC3b to assess phagocytosis by complement receptor 3. However, previous studies have shown phagocytosis of uncoated carboxylated beads by microglia. The authors should provide experimental evidence that phagocytosis is CR3 dependent in this assay.

Haptotaxis implies that biochemical cues are surface bound. While TIRF microscopy confirmed the formation of a gradient near the surface of the coverslip, this does not prove the absence of soluble iC3b. Additional controls would be necessary to ensure that cells only respond to surface-bound iC3b and not to soluble iC3b.

In the haptotaxis assays, the concentration of iC3b is lower in the area between the "cells" punch and the dish side than between the "cells" punch and the "iC3b" punch. The authors should comment on the possible implication on cell migration.

When drugs are added to cells under confinement, how is the final concentration calculated? Do the authors assume homogenous concentrations of drugs diffusing through the agarose?

Figure 2F is referred to twice, lines 163-164 and 197-198, with opposite conclusions. Statistical annotations on the figure suggest that the first statement is incorrect.

Line 24-250 "Arp2/3-deficient cells are capable of moving well under agarose". Please specify the cell type(s) and provide the reference(s).

Referee #3:

This manuscript by Paulson et al. presents an interesting investigation into the role of the Arp2/3 complex in microglial responses to the complement opsonin iC3b. The authors demonstrate that iC3b can act as an haptotactic cue for microglia. They show that Arp2/3 is required for both iC3b-related haptotaxis and phagocytosis, while it can be dispensable for migration towards other chemical cues, such as ATP, in microglia. The work elegantly employs innovative in vitro surface-coating and confinement assays as well as ex vivo imaging of murine hippocampal slices to study phagocytosis. Overall, this study is well-conducted, with important contributions to our understanding of microglial synaptic pruning and worth of publication to EMBO Reports. However, there are also some points that require improvement or clarification prior to publication.

1. In figures where multiple cells are quantified per condition but also the means are displayed, it is not clear how the statistics have been performed using the means or the individual data points. The use of very large n values (i.e., individual cells) can inflate statistical power and lead to significant p-values even in the absence of meaningful biological differences. It would be more informative and statistically appropriate to calculate statistics using the mean from each independent experiment rather than treating each cell as an independent data point.
2. In Figure 1D, the authors show uptake of AF-iC3b from the substrate, which is interpreted as phagocytosis. To confirm that this uptake is truly phagocytic and not due to general surface remodelling or endocytosis, it would be valuable to include a control substrate (e.g. fibronectin) and/or use specific phagocytosis inhibitors or CR3-blocking antibodies.
3. The authors can elaborate more on why they think that confinement increases bead uptake but not surface-bound phagocytosis. Also why some metrics are influenced by DMSO (e.g. 2F)?
4. The study is solely based on using the small molecule inhibitor CK-666 to inhibit the Arp2/3 complex. Have the authors considered using the small molecule inactive control for Arp2/3 inhibition CK-689 or silence the complex genetically to strengthen their argument that their results are specific to Arp2/3 complex? Also is the viability of cells either in vitro or ex vivo affected by CK-666?
5. Why CK-666 was polymerised in the agarose in Figure 3 and not given in the media? Is the drug bioavailability altered by agarose polymerisation?
6. Figure 3 can be improved by combining the 'away' and 'toward' cell track histograms in one graph for direct comparison of effect sizes. Also the authors can consider having the CK-666 data in the same graphs as the control also to facilitate direct comparisons.
7. The authors can expand their discussion to elaborate on whether there are iC3b gradients present in vivo. Are these altered in development or in pathophysiology?
8. Figures 5 and 6 would benefit from direct labelling of image panels with what each fluorescence channel represents (e.g., green: microglia (GFP); red: iC3b-beads). Additionally, in Figure 6, the red fluorescence signal corresponding to the beads appears markedly lower at the time of injection in the CK-666 condition compared to the control. Could the authors clarify whether this reflects a difference in bead injection efficiency, imaging exposure, or bead localization? Similarly, microglia in the CK-666 condition appear larger and more numerous. Is this due to altered morphology, a change in z-plane imaging, or a real difference in cell density or hypertrophy following Arp2/3 inhibition? Clarification of these differences would strengthen the interpretation of the data.
9. In their discussion the authors refer to trogocytosis, however, the presented evidence does not clearly distinguish trogocytosis from conventional phagocytosis. The authors should clarify the terminology and in general cautiously discuss their data, toning down arguments for therapeutic potential, unless additional evidence is available.

We thank the reviewers for their interest in our work, and for their thoughtful suggestions. We have taken their comments to heart and have addressed each concern below. Major, reviewer-suggested edits appear as red text in the revised manuscript.

Referee #1:

The paper is well-written, and the message is clear: microglial responses to iC3b occur in an Arp2/3 complex-dependent manner. Both iC3b-induced haptotaxis and phagocytosis of iC3b-coated beads are disrupted by Arp2/3 inhibitor treatment. This is further supported by the observation that the directional movement of microglia toward an iC3b gradient is abolished in the presence of CK-666. Additionally, the authors examine the role of Arp2/3 in microglia in situ, demonstrating that Arp2/3 contributes to maintaining microglial morphology and responsiveness to iC3b. This information advances our understanding of the molecular mechanisms regarding microglial haptosensing.

In my opinion there are only minor points need to be improved:

1. The label on the scale bar is too small to read, e.g., in Fig. 1D, 1J, 2E, 2J, 5A, 6A. Please review and correct this issue throughout the paper.

We thank the reviewer for bringing this to our attention. We have edited these figures to ensure that the scale bars are larger.

2. In line 335, the authors should rephrase the phrase 'it worth nothing...'. They can simply describe the observed trend, which is consistent with the pooled data. Moreover, the trends for both males and females appear similar. Statistical tests on these samples are unnecessary and uninformative, even misleading, as the sample sizes are too small to yield reliable conclusions. The same applies to Fig. 6E, F, J, K, and L.

We have removed the statistical analysis of sex-based differences, as suggested by the reviewer. We have removed these panels from the main figures and placed them in a new supplementary figure. In addition, we have revised the related results text in line with this reviewer's suggestion.

3. It has been reported that the sensitivity of the Arp2/3 complex to inhibitors can vary depending on its subunit composition. For example, ArpC1A-containing Arp2/3, which is more sensitive to CK-666, is predominantly found in neurons, whereas ArpC1B-containing Arp2/3, which can still generate actin branches in the presence of CK-666, is more common in immune cells. Do the authors have any information regarding the expression levels of Arp2/3 subunit isoforms in microglia? If so, it would be worthwhile to include this in the discussion.

This is an excellent and important point. In the revision we elaborate on this in the discussion, as suggested by the reviewer.

4. Fig 5A. I am not sure I understand what I'm looking at, and it's unclear what I expect to see. Additional explanation would likely help clarify this.

We apologize for the confusion regarding this figure. We have added additional explanation, as suggested by the reviewer. In addition, in the revision we zoom in on the region of interest so that readers will appreciate the gradual accumulation of green in the demarked zone indicating a response by nearby microglia to ATP.

Referee #2:

In this article, Paulson et al. Investigate the role of the Arp2/3 complex in the interaction of microglia with iC3b-bound surfaces or phagocytic targets. In the first part of the paper, they use a microglia-like cell line to characterize cell migration and phagocytosis in vitro. They show that these cells migrate on iC3b-coated surface, although slowly and less persistently than on fibronectin. Pharmacological inhibition of Arp2/3 reduces migration in both unconfined and confined conditions and prevents the detachment of iC3b from the surface. Furthermore, they observed that Arp2/3 inhibition reduced the phagocytosis of iC3b-coated beads, both in confined and unconfined conditions. Next, they look at the effect of a gradient of iC3b on microglia migration under confinement. Microglia migrated both towards and away from the gradient. However, their persistence was lower and velocity higher when migrating away from the iC3b gradient. These differences were abolished by Arp2/3 inhibition. Finally, they use brain tissue sections to investigate the migration of primary microglia in a physiological environment. Unsurprisingly, they found a role of Arp2/3 in the ability of microglia to form protrusions and maintain a ramified morphology. Yet, Arp2/3 inhibition had no significant effect in microglia chemotaxis towards ATP. However, microglia could not form stable interactions and internalize iC3b-coated beads when Arp2/3 was inhibited.

These results support the assertion that Arp2/3 plays an important role in iC3b phagocytosis by microglia, as previously demonstrated in macrophages by the senior author. This is likely due to a critical role of Arp2/3 in establishing strong binding to iC3b, as suggested by both in vitro and in situ experiments. However, several assertions made in the manuscript are not sufficiently supported by experimental evidence. The author claimed that phagocytosis under confinement in an Arp2/3 dependent manner, yet, when Arp2/3 is inhibited, phagocytosis appears to be in confined versus unconfined conditions. The authors also state that haptotactic migration toward increasing iC3b

concentration is Arp2/3 dependent. However, there is no statistical comparison of DMSO versus CK-666 treatment in Figure 3 and, as far as I can tell, Arp2/3 inhibition seems to only affect cells migrating away from the source of iC3b. Finally, the authors link the inability of microglia to bind to iC3b-coated beads in tissues to a haptotaxis defect. Since there is no gradient of iC3b on the beads, the experiment does not support this interpretation.

Additional comments:

In several figures, statistical analysis is not properly conducted. In general, statistics should compare various treatments or treated versus untreated cells within the same condition. For example, Figure S1C-E should compare the effect of different antibodies when for cells migrating on fibronectin on one hand, and cells migrating on iC3b on the other hand. This is critical for the interpretation of the data. In this instance, blocking beta2 does not seem to affect binding to iC3b. Likewise, in Figure 3, data from CK-666 treated cells should be compared to data from untreated cells.

We agree with the reviewer about Figures S1C-E. We have added statistical analysis demonstrating changes between control wells and the b2/aM-inhibited conditions. However, for Figure 3 there is not a good way to compare CK-666 treated cells in the under-agarose system to control, as the inhibitor is polymerized into the gel itself. However, we have included supporting data in the revision with the inactive control compound CK-689 and demonstrated that cells retain the ability to sense the iC3b gradient (new Supplementary Figure 3E). We feel that these experiments address the reviewer's concern about directly comparing Arp2/3-deficient cells to control in the same context.

Method descriptions lack clarity. In several places, the mention "as previously described" is not associated with any reference. In these cases, the authors should provide complete descriptions of their procedures. The description of the phagocytosis assay under confinement lacks clarity. Are beads added to the same punch? If so, how far from the punch images were acquired for quantification and what criteria were utilized to determine that distance?

We thank the reviewer for this feedback and agree that 'as previously described' was overused. In many cases we referred to methods described earlier in the section and have made this clear in the revision. In other cases, we have added more information and/or citations to help clarify where these items were previously described.

The authors use carboxylated beads coated with iC3b to assess phagocytosis by

complement receptor 3. However, previous studies have shown phagocytosis of uncoated carboxylated beads by microglia. The authors should provide experimental evidence that phagocytosis is CR3 dependent in this assay.

We thank the reviewer for this important point. In the revision we include data demonstrating that antibody inhibition of $\beta 2$ integrin impairs bead uptake compared to cells treated with normal IgG or untreated (Supplementary Figure S1G). In addition, we show that similar inhibition impairs uptake of iC3b from glass surfaces (Supplementary Figure S1F). We feel that together these data indicate a specific response to iC3b rather than to the bead material.

Haptotaxis implies that biochemical cues are surface bound. While TIRF microscopy confirmed the formation of a gradient near the surface of the coverslip, this does not prove the absence of soluble iC3b. Additional controls would be necessary to ensure that cells only respond to surface-bound iC3b and not to soluble iC3b.

We have included data in the revision that strongly suggests that there is little soluble iC3b in our system. We washed a series of dishes multiple times with 1X PBS after the gradient was initiated and found that the fluorescence intensity of iC3b along the gradient remained statistically similar after every wash (Supplementary Figure S3D). If there had been a large amount of soluble fluorescent iC3b, we would expect it to decrease as a function of the washes as it became progressively diluted over time.

In the haptotaxis assays, the concentration of iC3b is lower in the area between the "cells" punch and the dish side than between the "cells" punch and the "iC3b" punch. The authors should comment on the possible implication on cell migration.

We comment on this relationship extensively in our previous paper (Stinson, Paulson, et al. *Biology Open*, 2025). We have noticed that cells will move with a degree of persistence upon entering a confined setting, even against a gradient. However, cells move in a significantly more directional fashion when they are confined and moving in response to a gradient (e.g. iC3b in this case, or FN in our previous paper).

When drugs are added to cells under confinement, how is the final concentration calculated? Do the authors assume homogenous concentrations of drugs diffusing through the agarose?

Since drugs are added directly to the gel, it is possible to precisely control the concentration that goes into the under-agarose system. While it is difficult to know exactly how much of the drug is released in this context, our analysis in the present study demonstrates that the CK-666 in the gel is similarly effective compared to the expected concentration in media. Thus, we are comfortable taking these results at face

value and assuming that the working concentration of CK-666 in confinement is very similar to the concentration we put into the gel (which is equal to the dose given in unconfined conditions).

Figure 2F is referred to twice, lines 163-164 and 197-198, with opposite conclusions. Statistical annotations on the figure suggest that the first statement is incorrect.

We thank the reviewer for catching this typo. The first instance (lines 163-164) was supposed to refer to Figure 1F but was mistakenly referred to as Figure 2F.

Line 24-250 "Arp2/3-deficient cells are capable of moving well under agarose". Please specify the cell type(s) and provide the reference(s).

We have done as the reviewer has suggested. In addition to the current work on microglia, we include references to previous studies on macrophages and dendritic cells.

Referee #3:

This manuscript by Paulson et al. presents an interesting investigation into the role of the Arp2/3 complex in microglial responses to the complement opsonin iC3b. The authors demonstrate that iC3b can act as an haptotactic cue for microglia. They show that Arp2/3 is required for both iC3b-related haptotaxis and phagocytosis, while it can be dispensable for migration towards other chemical cues, such as ATP, in microglia. The work elegantly employs innovative in vitro surface-coating and confinement assays as well as ex vivo imaging of murine hippocampal slices to study phagocytosis. Overall, this study is well-conducted, with important contributions to our understanding of microglial synaptic pruning and worth of publication to EMBO Reports. However, there are also some points that require improvement or clarification prior to publication.

1. In figures where multiple cells are quantified per condition but also the means are displayed, it is not clear how the statistics have been performed using the means or the individual data points. The use of very large n values (i.e., individual cells) can inflate statistical power and lead to significant p-values even in the absence of meaningful biological differences. It would be more informative and statistically appropriate to calculate statistics using the mean from each independent experiment rather than treating each cell as an independent data point.

We appreciate the reviewer's point. While many of our analyses did consider only mean values (notably the hippocampal slice experiments), in other experiments the total number of sampled values was reported as the N. In these cases, we have taken the reviewer's comments to heart and have also included an appendix featuring effect size measurements. These effect sizes were calculated based on the mean of all experimental means (mean of means), which in many cases was an N of 3 or 4. It is our understanding that effect sizes are not as dependent on how many N are used in the experiment and give an indication about how much two populations differ from each other. It is also our understanding that size effects ≥ 0.8 are conventionally defined as having a high likelihood of difference between two populations. It is notable that many effect size values are substantially higher than 0.8 for statistically significant p-values reported in these experiments. We feel that including an effect size appendix alongside p-values strengthens the manuscript's conclusions.

2. In Figure 1D, the authors show uptake of AF-iC3b from the substrate, which is interpreted as phagocytosis. To confirm that this uptake is truly phagocytic and not due to general surface remodelling or endocytosis, it would be valuable to include a control substrate (e.g. fibronectin) and/or use specific phagocytosis inhibitors or CR3-blocking antibodies.

We thank the reviewer for making this suggestion. We have demonstrated in the past (Stinson et al., *MBoC*, 2024) that macrophages take up fibronectin during random migration, though we never tested whether the fibronectin was recycled or disposed of via phagolysosomal clearance. Considering this uncertainty, we decided to follow the reviewer's other suggestion to use CR3-blocking antibodies. We demonstrate that $\beta 2$ integrin inhibition blocks the ability of BV2 cells to take up the fluorescent iC3b (Supplemental Figure S1F). As stated in the revised manuscript, the effect of $\beta 2$ inhibition is likely more robust as it is expected to block both CR3- and CR4-mediated uptake.

3. The authors can elaborate more on why they think that confinement increases bead uptake but not surface-bound phagocytosis. Also why some metrics are influenced by DMSO (e.g. 2F)?

We thank the reviewer for making this suggestion. We do note that there is precedence for DMSO shifting the activity of sodium channels, which might affect their activation state. Text to this effect is presented right after we discuss this finding in the text. The surface uptake experiments were also done in the presence of DMSO, so this could also be a factor that explains the difference seen between data in 1F and 2D. The

molecular mechanism remains unclear, but previous studies have suggested that DMSO may also alter mouse behavior as well.

4. The study is solely based on using the small molecule inhibitor CK-666 to inhibit the Arp2/3 complex. Have the authors considered using the small molecule inactive control for Arp2/3 inhibition CK-689 or silence the complex genetically to strengthen their argument that their results are specific to Arp2/3 complex? Also is the viability of cells either *in vitro* or *ex vivo* affected by CK-666?

The viability of cells in both our *in vitro* and *in situ* experiments was not affected by CK-666, or by DMSO vehicle when delivered at the same volume as used in the CK-666 treatments. We have attempted to knock out the *Arpc2* allele of the Arp2/3 complex from microglia *in situ* using intraperitoneal delivery of tamoxifen in combination with the CX3CR1-CreER-IRES-YFP mouse allele. However, even control microglia (lacking the floxed *Arpc2* allele, or untreated with tamoxifen) in this context looked aberrant and could not be adequately harvested from neonatal or adult mice. We do not fully understand why this is the case, as we are able to routinely harvest neonatal microglia from C57Bl6/J control mice. For experiments where concurrent vehicle control sampling was not possible (i.e. haptotaxis experiments), we utilized the CK-689 negative control compound. Results with this treatment echoed the untreated condition, in that CK-689 did not affect the ability of BV2 cells to haptotax in the (+) condition, with an FMIx and persistence significantly higher than the (-) condition (Supplemental Figure S3E). These data increase our confidence that the CK-666 treatment specifically affects haptotactic sensing, and that the response to iC3b is Arp2/3 complex-dependent.

5. Why CK-666 was polymerised in the agarose in Figure 3 and not given in the media? Is the drug bioavailability altered by agarose polymerisation?

We do not know whether bioavailability is altered by agarose polymerization. However, we do demonstrate that CK-666 effectively suppressed Arp2/3 function when polymerized in the gel, as it acts similarly there compared to unconfined settings.

6. Figure 3 can be improved by combining the 'away' and 'toward' cell track histograms in one graph for direct comparison of effect sizes. Also the authors can consider having the CK-666 data in the same graphs as the control also to facilitate direct comparisons.

We thank the reviewer for this suggestion. We chose to leave the CK-666 data on a separate graph, as this was an independent experiment from the control experiment. We have taken the reviewer's advice about clarifying these data by showing a collection of cell tracks side by side in each condition rather than moving away from each other in

order to facilitate direct, qualitative comparison. In addition, calculation of Hedges' g value for size effect reveals a relatively large effect when comparing untreated cells in the (+) and (-) settings, with respect to FMIx (1.65) and persistence (1.16). On the other hand, comparison of the same conditions in the presence of CK-666 yielded very low effect sizes for FMIx (0.25) and persistence (0.099). It is our understanding that comparing size effects across experiments can be statistically meaningful (whereas comparing p-values across experiments may not be statistically robust). Therefore, direct comparison between these settings suggests that cells move more directionally toward an iC3b gradient, and that this specificity is lost upon Arp2/3 disruption.

7. The authors can expand their discussion to elaborate on whether there are iC3b gradients present in vivo. Are these altered in development or in pathophysiology?

We thank the reviewer for this excellent advice. This is a major, highly interesting point that we feel added a great deal to our discussion.

8. Figures 5 and 6 would benefit from direct labelling of image panels with what each fluorescence channel represents (e.g., green: microglia (GFP); red: iC3b-beads). Additionally, in Figure 6, the red fluorescence signal corresponding to the beads appears markedly lower at the time of injection in the CK-666 condition compared to the control. Could the authors clarify whether this reflects a difference in bead injection efficiency, imaging exposure, or bead localization? Similarly, microglia in the CK-666 condition appear larger and more numerous. Is this due to altered morphology, a change in z-plane imaging, or a real difference in cell density or hypertrophy following Arp2/3 inhibition? Clarification of these differences would strengthen the interpretation of the data.

We thank the reviewer for pointing this out. We have followed the reviewer's advice regarding coloration of image panels. With respect to the red fluorescence signal in Figure 6, we realized that the red levels in these images were not set the same during post-processing. We apologize for this oversight and have edited the figure accordingly. Having gone through our images again, we do not see any evidence that there is a change in microglia number upon CK-666 treatment across all the trials in this condition compared to DMSO control. We did find that the ratio of soma cell area to total cell area was altered by CK-666 treatment, though the raw values of soma size and total cell area were not significantly different (new Figure 4C). We interpret these data to mean that Arp2/3 inhibition causes a redistribution of cell mass to the soma at the expense of process extensions. This is further supported by the data demonstrating that CK-666 treated cells form fewer extensions and have a lower ramification index (Figure 4D). The difference in number of cells present in our example images likely reflects a minor

regional difference in microglia at this location rather than a biologically meaningful finding.

9. In their discussion the authors refer to trogocytosis, however, the presented evidence does not clearly distinguish trogocytosis from conventional phagocytosis. The authors should clarify the terminology and in general cautiously discuss their data, toning down arguments for therapeutic potential, unless additional evidence is available.

We thank the reviewer for this feedback. Our intention in the revision has been to note that there are many similarities between trogocytosis and phagocytosis. Microglial trogocytosis could be relevant to synaptic pruning, but it remains unclear exactly how similar the trogocytosis and phagocytosis pathways are. We have also clarified the definitions of these processes. In addition, we have toned down arguments for therapeutic potential. These were intended to be a 'future potential impact' rather than an interpretation of the present data. We also explicitly state that more work on this topic is required before we can fully understand the role of haptotaxis in neurodegeneration and whether it is a viable therapeutic target.

Dear Dr. Rotty,

Thank you for the submission of your revised manuscript to our editorial offices. I have already forwarded to you the reports I received from the three referees that I asked to re-evaluate your study. Please find them again below.

After going through your preliminary point-by-points response (further revision plan), I have decided to proceed. Please address the remaining concerns of referee #2 in a final revised manuscript and/or in a final detailed point-by-point response, as indicated in your revision plan. Moreover, I have several editorial requests. Please also provide a p-b-p-response regarding these with your final submission.

Editorial requests:

- Please reduce the number of keywords to five and order the manuscript sections like this, using only these names: Title page - Abstract (max. 175 words) - Keywords - Introduction - Results - Discussion - Methods - Data availability section - Acknowledgements (please include here all the funding information) - Disclosure and Competing Interests Statement - References - Figure legends - Expanded View Figure legends

- We now use CRediT to specify the contributions of each author in the journal submission system. CRediT replaces the author contribution section. Please use the free text box to provide more detailed descriptions and do NOT provide your final manuscript text file with an author contributions section. See also our guide to authors (section 'Author contributions'): <https://link.springer.com/journal/44319/submission-guidelines#cms-Revised-submissions>

- The data availability section is restricted to large datasets deposited at external repositories. If no datasets have been deposited, please state this here (e.g. 'No primary datasets have been generated and deposited') and remove all other text from this section.

- Please check again that the number "n" for how many independent experiments were performed, their nature (biological versus technical replicates), the bars and error bars (e.g. SEM, SD) and the test used to calculate p-values is indicated in the respective figure legends (main, EV and Appendix figures). Please also check that all the p-values are explained in the legend, and that these fit to those shown in the figure. Please provide statistical testing where applicable. Please avoid the phrase 'independent experiment' but clearly state if these were biological or technical replicates. Please also indicate (e.g. with n.s.) if testing was performed, but the differences are not significant. In case n=2, please show the data as separate datapoints without error bars and statistics. See also:

<https://link.springer.com/journal/44319/submission-guidelines#cms-Figure-and-data-presentation>

If $n < 5$, please show single datapoints for diagrams. Moreover:

- Please note that the exact p values are not provided in the legends of figures 1a-c, f, h, i; 2a-d, f, g, i; 3c-e; 4b, f; 6d-g: EV 1c, d, f, g; EV 3e

- Please indicate the statistical test used for data analysis in the legends of figures EV 1c-e

- Please note that information related to n is missing in the legends of figures 4c, d, f; EV 1c-g; EV 3a, b, d-g

- Please note that the error bars are not defined in the legends of figures 4c

- Please note that the measure of center for the error bars needs to be defined in the legends of figures EV 3e

- There is a file named 'Table - Appendix_Cohens d and Hedges g values' uploaded. Is this source data or a dataset? If this is a dataset, please upload it as Dataset EV1 and make sure it is called out like this. Please upload the original excel file and add a legend for on the first TAB.

- Please add scale bars of similar style and thickness to all microscopic images, using clearly visible black or white bars (depending on the background). Please place these in the lower right corner of the images themselves. Please do not write on or near the bars in the image but define the size in the respective figure legend. Presently, all scale bars have text nearby.

- Please make sure that all figure panels are called out separately and sequentially. Presently, it seems a callout for Fig. 2J is missing. Moreover, the callout 'Supplemental Figure 1F' has the wrong format. Should this be Figure EV1F? Please check.

- Please make sure that all the funding information is also entered into the online submission system and that it is complete and similar to the one in the acknowledgement section of the manuscript text file. Presently, the grants from the Uniformed Services University of the Health Sciences (USU), a Cosmos Club Foundation award, and startup funds from the Uniformed Services University, 20814-4799 are missing in the submission system. Please check.

- Please acknowledge BioRender with a paragraph (named Graphics) in the Methods section, indicating which panels/objects

were created using BioRender.com.

In addition, I would need from you uploaded separately:

I look forward to seeing the further revised version of your manuscript when it is ready.

Please let me know if you have questions regarding the revision.

Best,

Referee #1:

The authors have addressed all of my concerns. I believe the paper should be accepted and published in EMBO Reports.

Referee #2:

This submission included multiple revisions that improved the manuscript. However, as pointed out in my previous review, the fundamental issue with this paper is a significant disconnect between major assertions made by the authors and the results they obtained. In its current form, the interpretation of the data is misleading and should not be published as it is. Unfortunately, the authors did not address my main concerns in their revised version and only responded to my additional comments instead. Thus, I will reiterate my concerns regarding their statements here.

Statement #1: "With iC3b-opsionized beads, phagocytosis increased under confinement, again in an Arp2/3 complex-dependent fashion." Figure 1F shows that when Arp2/3 is inhibited by CK-666 (right side), confinement increases phagocytosis significantly. Thus, the increase of phagocytic activity induced by confinement occurs in an Arp2/3 independent manner. In fact, this increase is more significant in CK-666 treated cells than in DMSO treated cells. So, the conclusion made by the authors is in contradiction with their data.

Statement #2: "Likewise, when these cells were tasked with moving under confinement on iC3b in the presence of CK-666, we see a decrease in migratory abilities including loss of haptotaxis and a decrease in phagocytosis" This conclusion can not be made without statistical comparisons between DMSO and CK-666 treated cells. In the current Figure 3, these conditions are presented completely separately. To make a point regarding the effect of CK-666, Figure 3C should be compared to Figure 3I, Figure 3D should be compared to Figure 3J, Figure 3E should be compared to Figure 3K. Moreover, the only noticeable differences are when cells migrate in absence of iC3b gradient. Thus, the data suggests that Arp2/3 plays a role in random migration but has no effect on haptotaxis. Here again, the author's assertion is not in contradiction with the data.

Statement #3: "Together our data suggests that microglia haptotaxis is Arp2/3-dependent in an in vivo-like environment." Figure 6 suggests that binding to iC3b-beads is effected by CK-666 in vitro. All the beads have the same fluorescence intensity, so it is unclear whether none or all those beads have been phagocytosed. Regardless, there is no evidence of iC3b gradient in these in vivo experiments. So, the authors could conclude that Arp2/3 plays a role in migration and bead binding in vivo, but they can not make conclusions on haptotaxis based on these experiments.

Minor comments:

Line 151-152 "The 2-micron bead size was chosen due to its similarity to the size of iC3b-tagged dendritic spines internalized by microglia during synaptic pruning" are cited. Please provide references for the measurements of dendritic spine sizes.

For figure 6A, line 346-347. The authors claimed that DMSO-treated microglia interact and pull iC3b beads out of the ROI. Neither time lapse images nor videos are showing this convincingly. I would suggest revising this statement.

The authors should comment on why the accumulated distance and velocity was still significantly higher during CK-666

inhibition for the in situ experiments (figure 6), whereas in figure 2 microglia displayed significantly less velocity, accumulated distance and persistence under confined conditions, which should be the experimental model of the physiological condition. In many of the figures, $n=50$ cells per experiment is used over several replicates. Although both individual data points and mean of each replicate is usually present, it is unclear which n (number of cells or replicates) was used for statistical analysis. If each measured cell point is used, this can skew the statistical power of the analysis. Likewise Figure EV1 F has statistical analysis between two replicates, addition of at least one more replicate would strengthen the reliability of this experiment.

Referee #3:

The authors adequately addressed all my comments and I'm happy to recommend their work for publication in EMBO reports without further revisions.

We thank all reviewers for their excellent suggestions and support for this manuscript. We feel that the newest version is much improved by the reviewers' comments. As reviewers 1 and 3 recommended acceptance, we have focused the bulk of this response on reviewer 2's outstanding concerns.

This submission included multiple revisions that improved the manuscript. However, as pointed out in my previous review, the fundamental issue with this paper is a significant disconnect between major assertions made by the authors and the results they obtained. In its current form, the interpretation of the data is misleading and should not be published as it is. Unfortunately, the authors did not address my main concerns in their revised version and only responded to my additional comments instead. Thus, I will reiterate my concerns regarding their statements here.

We thank the reviewer for their attention to detail, and for reiterating the specific concerns underlying their review. We apologize for not tackling these issues explicitly in our first response and provide text edits to many of these objections that we feel brings each statement more in line with reviewer concerns.

Statement #1: "With iC3b-opsonized beads, phagocytosis increased under confinement, again in an Arp2/3 complex-dependent fashion." Figure 1F shows that when Arp2/3 is inhibited by CK-666 (right side), confinement increases phagocytosis significantly. Thus, the increase of phagocytic activity induced by confinement occurs in an Arp2/3 independent manner. In fact, this increase is more significant in CK-666 treated cells than in DMSO treated cells. So, the conclusion made by the authors is in contradiction with their data.

The data as presented in Figure 2F indicates that CK-666 significantly decreases phagocytosis when cells are unconfined (conditions 1 and 3 in this graph) and when they are confined (conditions 2 and 4). We interpret these data to mean that phagocytosis is indeed inhibited in both contexts when Arp2/3 is impaired. The question of whether confinement can partially compensate for Arp2/3 loss is interesting, but outside the scope of the present paper.

Statement #2: "Likewise, when these cells were tasked with moving under confinement on iC3b in the presence of CK-666, we see a decrease in migratory abilities including loss of haptotaxis and a decrease in phagocytosis" This conclusion can not be made without statistical comparisons between DMSO and CK-666 treated cells. In the current Figure 3, these conditions are presented completely separately. To make a point regarding the effect of CK-666, Figure 3C should be compared to Figure 3I, Figure 3D should be compared to Figure 3J, Figure 3E should be compared to Figure 3K. Moreover, the only noticeable differences are when cells migrate in absence of iC3b gradient. Thus, the data suggests that Arp2/3 plays a role in random migration but has no effect on haptotaxis. Here again, the author's assertion is not in contradiction with the data.

This is a similar comment to one which we addressed from reviewer 3 in our response to reviewers (which follows in red text). The experiments in Figure 3C-F and Figure 3I-L were different experiments, due to the nature of the under-agarose assay. As these are different experiments, we feel it is inappropriate to compare them with p-values. Instead, it is possible to compare effect sizes between groups, and we have done so, as suggested by the reviewer (these were included with the resubmission as supplementary material): In addition, calculation of Hedges' g value for size effect reveals a relatively large effect when comparing untreated cells in the (+) and (-) settings, with respect to FMIx (1.65) and persistence (1.16). On the other hand, comparison of the same conditions in the presence of CK-666 yielded very low effect sizes for FMIx (0.25) and persistence (0.099). It is our understanding that comparing size effects across experiments can be statistically meaningful (whereas comparing p-values across experiments may not be statistically robust). Therefore, direct comparison between these settings suggests that cells move more directionally toward an iC3b gradient, and that this specificity is lost upon Arp2/3 disruption.

We have added the text below to our results section detailing the Figure 3 data to help clarify our interpretation for readers in two separate results paragraphs (red text is newly added):

We also calculated effect sizes using Hedges' g and found values of 1.65 for FMI and 1.15 for persistence when comparing the (+) and (-) populations (Table EV1). Cells moving along the iC3b gradient (+) are slower compared to cells in the (-) orientation but showed no change in total distance (Figure 3E-F). These analyses together suggest that cells moving toward the iC3b gradient do so with higher directionality than cells moving away from the gradient, which is consistent with haptotactic migration.

Hedges' g values suggest that the effect size was low when comparing (+) and (-) CK-666 treated populations for both FMI (0.25) and persistence values (0.099). These analyses together suggest that Arp2/3 complex disruption causes cells to no longer prefer the (+) condition over the (-) condition, which is consistent with loss of haptosensing.

Statement #3: "Together our data suggests that microglia haptotaxis is Arp2/3-dependent in an in vivo-like environment." Figure 6 suggests that binding to iC3b-beads is effected by CK-666 in vitro. All the beads have the same fluorescence intensity, so it is unclear whether none or all those beads have been phagocytosed. Regardless, there is no evidence of iC3b gradient in these in vivo experiments. So, the authors could conclude that Arp2/3 plays a role in migration and bead binding in vivo, but they can not make conclusions on haptotaxis based on these experiments.

As measured in Figure 6D, CK-666 treated microglia interact with fewer beads despite forming processes that are similar in size (Figure 6C) and highly mobile (Figure 6E). However, they do not move persistently in the presence of beads like DMSO treated cells do (Figure 6G). Our assay was not aimed at measuring phagocytosis of beads, which is why the beads retain the same fluorescence intensity throughout the experiment. The data presented here do suggest that CK-666 treated microglia fail to sense beads in their proximity (Figure 6D) and that they cannot persistently move their processes in a coherent fashion to interact with beads (Figure 6G). Though we cannot speak to an *in vivo* gradient of iC3b, we do feel that these responses (sensing and coherent response to an immobilized cue) echo a haptotactic response.

Nonetheless, we will amend this statement as follows: “Together our data from an *in vivo*-like environment suggest that Arp2/3 complex-dependent haptosensing could be relevant for microglial clearance of iC3b-labeled targets in the CNS.”

Minor comments:

Line 151-152 "The 2-micron bead size was chosen due to its similarity to the size of iC3b-tagged dendritic spines internalized by microglia during synaptic pruning" are cited. Please provide references for the measurements of dendritic spine sizes.

We will clarify this statement and amend it to “The 2-micron bead size was chosen due to its similarity to the range of spine lengths measured experimentally” (Citations: Nakayama et al., J. Neurosci 2000; Benavides-Piccione et al., J Comp. Neurol. 2025)

For figure 6A, line 346-347. The authors claimed that DMSO-treated microglia interact and pull iC3b beads out of the ROI. Neither time lapse images nor videos are showing this convincingly. I would suggest revising this statement.

We will amend this from “Remarkably, DMSO-treated microglia stably interact with iC3b beads and end up consolidating and pulling them out of the ROI over time” to “Remarkably, DMSO-treated microglia stably interact with iC3b beads, and consolidate them over time.”

The authors should comment on why the accumulated distance and velocity was still significantly higher during CK-666 inhibition for the *in situ* experiments (figure 6), whereas in figure 2 microglia displayed significantly less velocity, accumulated distance and persistence under confined conditions, which should be the experimental model of the physiological condition.

This could be due to differences in how the measurements were taken, or differences in cells used in the study (BV2 cell line *in vitro*, primary cells *in situ*). Measurements in Figure 2 were taken from whole cell migration, whereas Figure 6 measurements were taken of protrusions moving toward the injection site. Microglia do not undergo whole cell migration *in situ* over the timeframe

measured here. Finally, the morphology of BV2s is quite different than the morphology of primary cells *in situ*, which could give rise to different responses in these contexts.

In many of the figures, n=50 cells per experiment is used over several replicates. Although both individual data points and mean of each replicate is usually present, it is unclear which n (number of cells or replicates) was used for statistical analysis. If each measured cell point is used, this can skew the statistical power of the analysis. Likewise Figure EV1 F has statistical analysis between two replicates, addition of at least one more replicate would strengthen the reliability of this experiment.

Reviewer 3 had a similar concern, which we addressed in the previous round of revision: We appreciate the reviewer's point. While many of our analyses did consider only mean values (notably the hippocampal slice experiments), in other experiments the total number of sampled values was reported as the N. In these cases, we have taken the reviewer's comments to heart and have also included an appendix featuring effect size measurements. These effect sizes were calculated based on the mean of all experimental means (mean of means), which in many cases was an N of 3 or 4. It is our understanding that effect sizes are not as dependent on how many N are used in the experiment and give an indication about how much two populations differ from each other. It is also our understanding that size effects ≥ 0.8 are conventionally defined as having a high likelihood of difference between two populations. It is notable that many effect size values are substantially higher than 0.8 for statistically significant p-values reported in these experiments. We feel that including an effect size appendix alongside p-values strengthens the manuscript's conclusions.

Dr. Jeremy Rotty
Uniformed Services University of the Health Sciences
Biochemistry
Bethesda
United States

Dear Dr. Rotty,

Thank you for the submission of your final revised manuscript to our editorial offices. I now went through it and your final p-b-p-response and consider the remaining concerns and suggestions of referee #2, and the editorial requests, as adequately addressed.

I am thus very pleased to accept your manuscript for publication in the next available issue of EMBO reports. Thank you for your contribution to our journal.

You may qualify for financial assistance for your publication charges - either via a Springer Nature fully open access agreement or an EMBO initiative. Check your eligibility: <https://link.springer.com/journal/44319/how-to-publish-with-us>

Yours sincerely,

>>> Please note that it is EMBO Reports policy for the transcript of the editorial process (containing referee reports and your response letter) to be published as an online supplement to each paper. If you do NOT want this, you will need to inform the Editorial Office via email immediately. More information is available here: <https://link.springer.com/partners/embo-press/editorial-policies#Peer%20review>